# Predicting natural variation in the yeast phenotypic landscape with machine learning

Sakshi Khaiwal[1]✉, Matteo De Chiara[1], Benjamin P Barré[1], Inigo Barrio-Hernandez [ID][2], Simon Stenberg[3], Pedro Beltrao [ID][2], Jonas Warringer [ID][3] & Gianni Liti [ID][1]✉

## Abstract

**Most organismal traits result from the complex interplay of many genetic and environmental factors, making their prediction difficult. Here, we used machine learning (ML) models to explore phenotype predictions for 223 traits measured across 1011 genome-sequenced *Saccharomyces cerevisiae* strains isolated worldwide. We benchmarked a ML pipeline with multiple linear and non-linear models to predict phenotypes from genotypes and gene expression, and determined gradient boosting machines as the best-performing model. Gene function disruption scores and gene presence/absence emerged as best predictors, suggesting a considerable contribution of the accessory genome in controlling phenotypes. The prediction accuracy broadly varied among phenotypes, with stress resistance being easier to predict compared to growth across nutrients. ML identified relevant genomic features linked to phenotypes, including high-impact variants with established relationships to phenotypes, despite these being rare in the population. Near-perfect accuracies were achieved when other phenomics data mostly in similar conditions were used, suggesting that useful information can be conveyed across phenotypes. Overall, our study underscores the power of ML to interpret the functional outcome of genetic variants.**

**Keywords** Machine Learning; Prediction; Genetic Variants; Phenotypes; *S. cerevisiae*
**Subject Categories** Genetics, Gene Therapy & Genetic Disease; Microbiology, Virology & Host Pathogen Interaction

## Introduction

The majority of traits are complex and controlled by multiple genetic loci, the environment, and interactions between the two. This interplay of many variables makes it difficult to determine their underlying genetic architecture (Goddard et al, 2016; Ehrenreich et al, 2010). Genome-wide association studies (GWAS) have proven successful in associating genetic markers to traits in species for which large cohorts can be sequenced and phenotyped (Zhang et al, 2020; de Lange et al, 2017; Li et al, 2017; Tian et al, 2020). However, small effects and rare variants typically fail to pass the GWAS stringent statistical thresholds. For combinations of variants, the problem becomes exponentially bigger, and only the strongest effects of very common pairs can be detected. Therefore, we only have a partial view of the genetic architecture even for well-studied traits (Tam et al, 2019).

Machine learning (ML) is emerging as a powerful set of tools capable of constructing models with the potential to capture even complex genotype–phenotype relationships and to enable predictions (Guo and Li, 2023) by identifying and prioritizing causative variants (Lakiotaki et al, 2023; Opulente et al, 2024; Harrison et al, 2024; Gonçalves et al, 2024). These models can be further powered by including additional information, such as multi-omics phenotypes (Schrag et al, 2018; Wang et al, 2024) and evolutionary frameworks (Cheng et al, 2021). However, both GWAS and predictions often lose power when environmental factors cannot be controlled (Guo and Li, 2023). Model organisms such as *Saccharomyces cerevisiae* are powerful systems for investigating these genotype–phenotype relationships due to the relative ease with which accurate genotypic and phenotypic information in controlled environments can be obtained (Yeh et al, 2022; Liti et al, 2017; Fay, 2013; Liti and Louis, 2012). The genome sequenced 1011 *S. cerevisiae* collection, isolated worldwide from a broad variety of ecological niches and domesticated environments, provided an accurate population-level variation portrait (Peter et al, 2018). An ensemble of life history phenotypes measured under controlled environmental conditions captures core components of the species' life cycle such as growth, sporulation, survival under starvation conditions, and cell characteristics such as cell size, DNA content, and mitochondrial activity (Peter et al, 2018; De Chiara et al, 2022; D'Angiolo et al, 2023; Galardini et al, 2019). Both proteome and transcriptome of the 1011 *S. cerevisiae* collection have also been quantified, providing molecular traits that can aid predictions of higher-level phenotypes from genetic variants (Caudal et al, 2024; Muenzner et al, 2024). Furthermore, investigation of *S. cerevisiae* natural variation can be coupled with multiplexed genome editing approaches, providing a platform to test some standard state-of-the-art ML predictions at scale (Sharon et al, 2018; Roy et al, 2018). While ML successfully predicted binary growth phenotypes across >1000 *Saccharomycotina* yeast species (Harrison et al, 2024;

[1]CNRS, INSERM, IRCAN, Côte d'Azur University, Nice, France. [2]Institute of Molecular Systems Biology, ETH Zürich, Zürich 8093, Switzerland. [3]Department of Chemistry and Molecular Biology, University of Gothenburg, Gothenburg 40530, Sweden. ✉E-mail: sakshi.KHAIWAL@univ-cotedazur.fr; gianni.liti@cnrs.fr

Gonçalves et al, 2024; Harrison et al, 2025), an in-depth investigation of whether these approaches enable predictions at a quantitative scale remains an open question. Here, we systematically explored four ML algorithms for phenotypic predictions in the 1011 *S. cerevisiae* collection across 223 phenotypes. To construct genotype–phenotype models that can enable predictions, we built Gen-phen, a flexible pipeline to quantify the best ML model and type of input information for predicting phenotypes and to build prediction models across the genotype and phenotype space of the *S. cerevisiae* collection.

## Results

### The *S. cerevisiae* natural phenotypic landscape

To define the phenotypic landscape of baker's yeast, we compiled 190 published life history traits measured across the 1011 *S. cerevisiae* collection in four main studies (Peter et al, 2018; De Chiara et al, 2022; D'Angiolo et al, 2023; Galardini et al, 2019) and complemented these with 33 additional traits (Dataset EV1). The aggregated phenome dataset comprises 223 phenotypes largely corresponding to yeast life cycle traits, including chronological life span (CLS), sporulation efficiency, and asexual reproduction (cell yield and doubling time) (Fig. 1A). Traits were measured across a range of controlled environments, including variations in type and concentration of carbon and nitrogen source availability and in exposure to drugs and abiotic stresses, such as temperature and salinity. We classified this ensemble of phenotypes into eight main classes, based on trait type or trait type-condition pairs (Fig. 1A,B; Dataset EV1). We first explored patterns of co-variation between traits, through an all-vs.-all pairwise correlation analysis and found significant correlations to be both more abundant and stronger than anticorrelations (Pearson's $R > 0.5$ for 6.5% of pairs vs. $R < -0.3$ for 0.1%) (Fig. 1C). Small, strongly correlated clusters are mostly driven by similar environments, or different concentrations of the same condition (Fig. 1C). Overall, fundamental trait types, such as CLS, growth yield, growth rate etc. had a stronger impact than environments (e.g., carbon and nitrogen sources, chemicals etc.) on phenotypic correlations. Thus, phenotypes of the same trait type measured across environments grouped together (Fig. 1C). However, 13% of the correlations and 60% of anticorrelations emerged between phenotypes of different classes. For example, cell yield during growth in the presence of azoles, which block the formation of mature sterols (Draskau and Svingen, 2022), positively correlated with CLS (Fig. 1C). Anticorrelations ($R < -0.3$) between cell growth rate and yield within the same environment, were more common than for other pairs of phenotypes (1.6% vs. 0.6%), but still rare, consistent with such trade-offs manifesting only under specific environments (Wei and Zhang, 2019).

Next, we constructed phenotypic networks based on strongly correlated phenotypes ("Methods"). Consistently, we observed major trait types, such as growth rate, yield, and CLS forming distinct subnetworks (Fig. 1D). Similar phenotypes measured in different studies were connected, as highlighted by the case of cycloheximide (CHX) (starred, Fig. 1D). Twelve phenotypes did not show strong correlation with other traits and remained unconnected (Fig. 1D). The low, or no, connectedness of these phenotypes is not due to higher experimental noise (Dataset EV2)

and instead suggests they are controlled by distinct biological pathways. Networks of anticorrelations were substantially less frequent and weaker compared to correlations with phenotypes being disconnected, except for six small subnetworks (Appendix Fig. S1A). Three of these subnetworks represent trade-offs between growth rate and yield in several sugars and drugs. To group the phenotypes based on correlation matrix, we performed hierarchical clustering and cut the resulting tree at $k = 4$, which was determined to be the optimal number of clusters based on Silhouette's score (calculated using k-means clustering, with k ranging from 1 to 20). The hierarchical groupings were able to recapitulate major patterns observed in the correlation map (Appendix Fig. S1B). For example, CLS clustered together with azole resistance, and growth yield across conditions from the three independent datasets clustered together.

To corroborate the above findings, we compared the natural phenotypic correlation patterns with correlations obtained from the yeast deletion collection, which consists of ~4800 single non-essential gene deletions derived from the *S. cerevisiae* reference laboratory strain background (Giaever et al, 2002). We retrieved 200 phenotypes measured across the yeast deletion collection and strongly overlapping with our phenotype dataset (Turco et al, 2023) (Dataset EV3). Clustering analysis on the correlation matrix of the gene-deletion phenotypes supported instances of shared correlation with what was observed in the natural 1011 collection, including drug resistance and CLS (Appendix Fig. S2A). Thus, these are likely true biological effects rather than spurious correlations due to shared bias.

Next, we investigated to what extent the co-variation in phenotypes agreed with overall genetic relatedness, and whether co-variation in specific phenotypes was driven by some specific phylogenetic clades. Strains in a few clades, such as French Guiana, Far East Asia, and Asian fermentation, were phenotypically similar (Fig. 1E). These clades showed very low intra-clade genetic variance, consistent with a higher degree of phenomic similarity (Appendix Fig. S2B–D). Overall, comparing phenomic and genetic distances, either based on SNPs (Pearson's correlation = 0.1, *p-value* = 0) or gene presence-absence (Pearson's correlation = 0.09, *p-value* = 0), we found only a very weak correlation. Thus, although very closely related yeasts show similar phenotypes, population structure across larger evolutionary distance, and phylogeny in general, appear to have little influence on *S. cerevisiae* intra-species phenotypic variation. Instead, wild and domesticated yeast clades clustered separately, despite different yeast clades having been domesticated independently at different time points and geographic regions. This is consistent with previous analysis (De Chiara et al, 2022; Tengölics et al, 2024), and supports that domestication is the major determinant of yeast phenotypic variation.

### The yeast GWAS catalog

To set a baseline for evaluating the capacity of ML to explore the yeast genotype–phenotype map, we first established genotype–phenotype connections using state-of-the-art GWAS approaches. We obtained a catalog of 2312 SNPs, distributed across 1271 genes (Dataset EV4), associating to one or more phenotypes (Fig. 2A). In addition, 34 loss-of-function variants (LOFs) in distinct genes, 65 gene presence/absence variations in the pangenome (PA) and 139 copy number variants (CNVs) associated

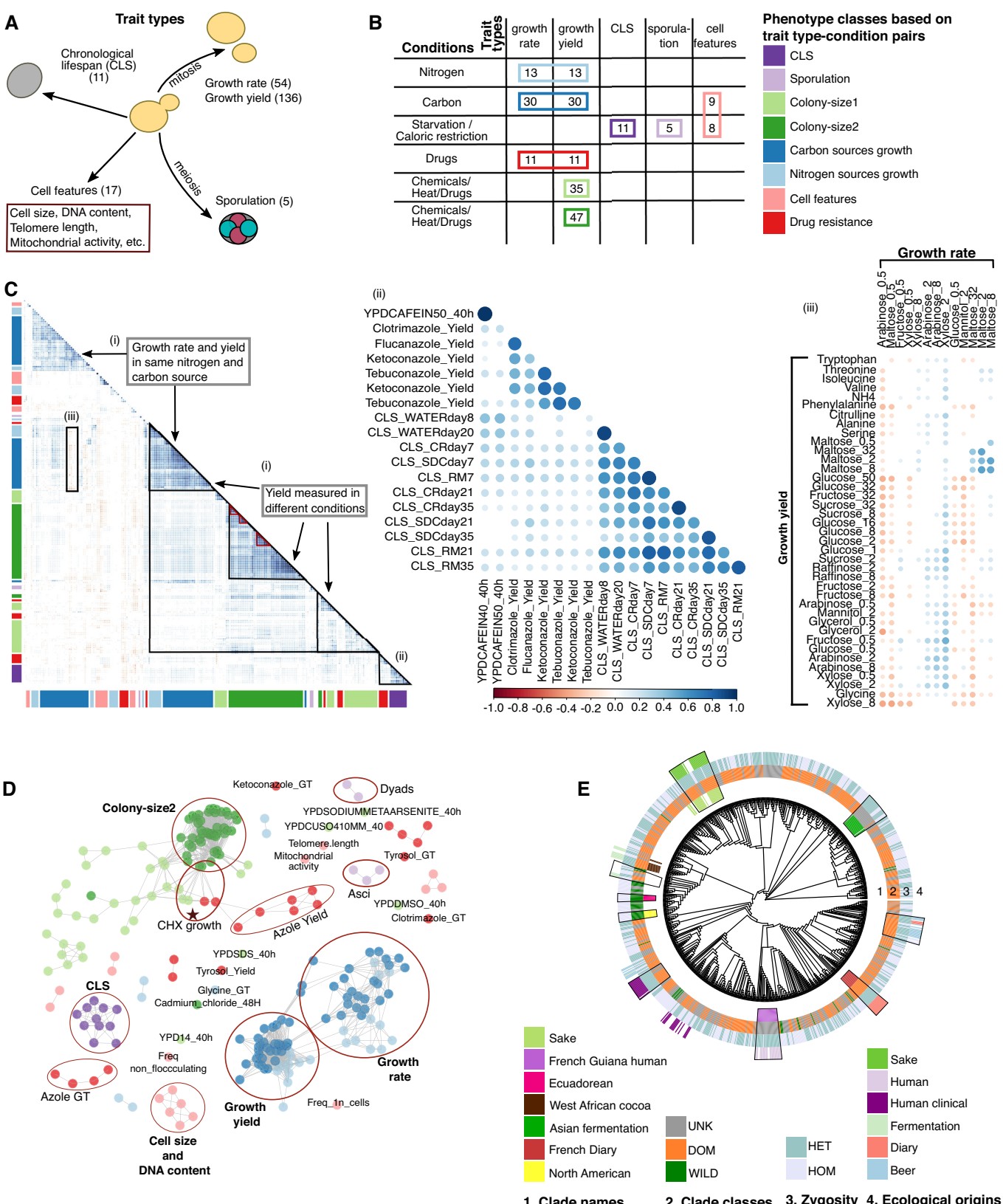

**Figure 1. The phenotypic landscape.**

(A) Schematic of measured phenotypes corresponding to key features of the yeast life cycle and cells. (B) Phenotype classes based on trait-type and environmental conditions. Rectangles colors around the numbers depict the phenotype classes (listed in the right). (C) Global phenotypic correlation map representing Pearson's correlation/anticorrelation and strength (respectively, dots colors and sizes). Non-significant correlations/anticorrelations (FDR corrected p-value > =0.05) are depicted by blank spaces. Color bars on the x and y axes indicate phenotype classes. Small clusters show examples of strong correlations between phenotypes in similar conditions (red triangles). Larger correlation patterns (black lines) show (i) clustering of similar traits (yield) across environments, (ii) unexpected significant correlations between unrelated traits (Chronological life span (CLS) and drug resistance), (iii) trade-offs between growth rate and yield. (D) Phenotypic networks based on correlation (>=0.5) reveal compact clusters within traits/environments and few disconnected phenotypes. Dots color indicates phenotype classes. The highlighted cluster by star depicts the connections between growth measured in cycloheximide (CHX) in three different studies. (E) Phenomic tree of the 1011 *S. cerevisiae* circled by heat maps indicating clades, clade classes (unknown (UNK), domesticated (DOM) and wild), zygosity (homozygous (HOM) and heterozygous (HET)), and ecological origins. Clusters overlapping phylogenetic clades are highlighted (black lines).

with at least one trait. While other variants associated proportionally with all phenotype classes, CNVs were enriched in cell features, CLS and colony-size 1 and 2 (Appendix Fig. S3A). Generally, the strength of phenotype associations, reflecting effect sizes, were lower for SNP than for PA, CNVs and LOF variants (Appendix Fig. S3B). The phenotype-associated variants were uniformly distributed across the genome, except for a strong enrichment associated with colony sizes 1 and 2 (from 34% genome-wide to 90%), within a chromosome XV region (Fig. 2A and Appendix Fig. S3C). These variants mostly mapped to the stress response transcriptional activator *SFL1* and the adjacent actin-related chromatin remodeller *ARP8* (Baccarini et al, 2015) (5 and 9 variants, associating with 18 and 20 phenotypes, respectively).

Three SNPs had exceptionally high significant associations (Fig. 2A), including a promoter variant (IME1_C-181T) in the master regulator *IME1* leading to sporulation defect (De Chiara et al, 2022) (Fig. EV1A), *AIM29* (C159T, His53His) causing azole resistance, (Fig. EV1B), and *SKN7* promoter variant (SKN7_C-376G) within a highly conserved region associated to ketoconazole resistance (Fig. EV1C). We used a variant effect predictor (VEP) to annotate all phenotype-associated variants. Surprisingly, synonymous (32%), missense (29%), and intergenic (37%) were equally represented among our GWAS hits (Fig. 2B), with fractions closely resembling how common these variant types are overall (Appendix Fig. S3D). Furthermore, the strength of associations was similar for these variant categories (Fig. 2B). Using the mutfunc framework (Wagih et al, 2018), we found no enrichment of variants predicted to affect function or protein stability among phenotype associations, whereas protein interfaces had a low number of predictions available (Dataset EV4; Appendix Fig. S3E). Phenotype-associated variants also affected essential and non-essential genes equally often (Appendix Fig. S4A).

We calculated the fraction of GWAS hits that was shared between each pair of phenotypes (Fig. 2C) and found both correlated and anticorrelated phenotypes to share variants more often than expected by chance (Fig. 2C, red boxes), supporting that these relations have a genetic rather than an environmental (i.e., shared bias) basis (Appendix Fig. S5B). Consistent with their strong correlations, cell yields across environments often shared GWAS hits (Fig. 2C, black boxes), while the same variant sharing was not evident for correlated CLS and azole resistance phenotypes. Expanding this analysis to the gene-level by associating every polymorphic position to the corresponding gene (Appendix Fig. S5C), we however found variants in the *FKS1* (Hu et al, 2023) gene to associate with both CLS and azoles resistance, indicating defects in cell wall biosynthesis as a potential basis of

their phenotypic correlation. We also identified three synonymous variants in the *SEC11* gene associated with the growth rate and yield trade-off in some sugar conditions, but swapping *SEC11* alleles between high and low-performing strains provided very variable results depending on the genetic-background with no conclusive support for causal effects (Dataset EV5), which could imply confounding intra or interlocus interactions.

Counting the number of phenotype associations per gene, we found 40% of the GWAS-associated genes (519 out of 1271) to be pleiotropic, with phenotypes reflecting cell-environment interactions, e.g., binding processes, cellular response to stimuli, and signal transduction, being enriched among GWAS hits (Dataset EV6; Appendix Fig. S4D). Often, a few SNPs within these genes accounted for almost all of the pleiotropy, e.g., 2 SNPs in *CAC2* and 3 SNPs in *CDC23* associating with 28 and 24 phenotypes, respectively, while a single SNP in the functionally uncharacterized *YEL023C* gene associated with 10 phenotypes.

We used a Personalized Page Rank (PPR) algorithm that tracks signal expansion through an interactome to identify pleiotropic gene groups, using both our GWAS hits and published phenotype data on the yeast gene knockout collection as inputs. To identify pleiotropic gene modules that impact multiple complex traits, we extracted modules with more than 75% genes in common between the two traits (Appendix Fig. S6A,B). For both the natural strain and gene knockout collections, only a few such modules were identified between similar phenotypes within each collection individually; however, no overlap was found between similar phenotypes across the two collections (Appendix Fig. S6A,B; Dataset EV7). For the natural collection, modules overlapping in multiple traits were mostly related to stress environments with enrichment for ubiquitin-like protein transferase activity and protein binding, while modules associated with sugar conditions were enriched in actin and cytoskeletal protein binding and other basal cell functions. However, few modules enriched for nucleic acid binding overlap between sugar conditions and stress environments (Dataset EV7).

## An integrated machine-learning pipeline for yeast phenotype prediction

The availability of 223 phenotypes coupled with genetic features (SNPs, LOF, PA, CNVs, P(AF) scores, detailed in "Methods"), and molecular-level (transcriptomic and proteomic) variations enabled us to predict *S. cerevisiae* phenotypes. We developed an all-in-one flexible pipeline (Gen-phen) for data pre-processing, feature selection and model learning to automate phenotype predictions

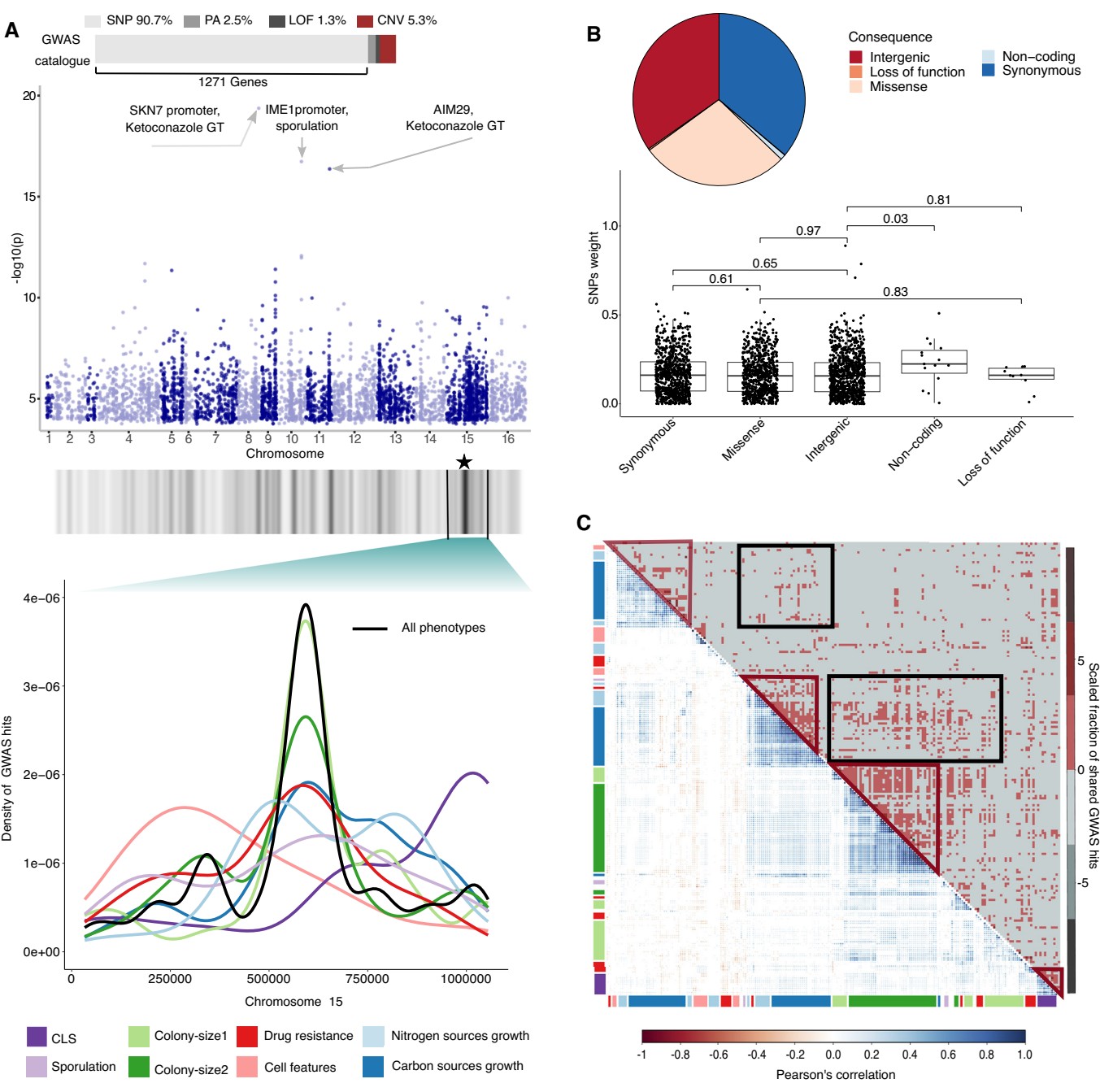

**Figure 2. GWAS overview.**

(**A**) Barplot (top) depicts the GWAS hits catalog with SNPs constituting 90% of hits in 1271 genes, followed by CNVs (5.3%), PA (2.5%), and LOF(1.3%). Genome-wide distribution of significant GWAS hits for the global phenome set (*P* values are significance values obtained from linear mixed models). The three top-associated variants in *SKN7*, *IME1*, and *AIM29* are annotated (*p-values* < e-16). The region with the highest hit density (box) in chromosome XV is zoomed-in (bottom panel), and line colors indicate phenotype classes. Colony sizes 1 and 2 mostly consist of cell yield in stressful environments and are highly enriched within this region. (**B**) Fractions of GWAS hits by type (pie chart) and weight distributions (box plot with Wilcoxon test *p-values*). The number of points in each box is given in Dataset EV2. The median (50th percentile) of the data is indicated by the black horizontal line, the lower and upper bound of the box indicate the 25th (Q1) and 75th (Q3) percentile, respectively. The whiskers (vertical lines) extends to the smallest and largest value within 1.5* interquartile range (Q3-Q1), and the points beyond the ends of the line are potential outliers. (**C**) Comparison between the phenomic correlation map (bottom left) and the scaled fraction of GWAS hits shared between phenotypes (top right).

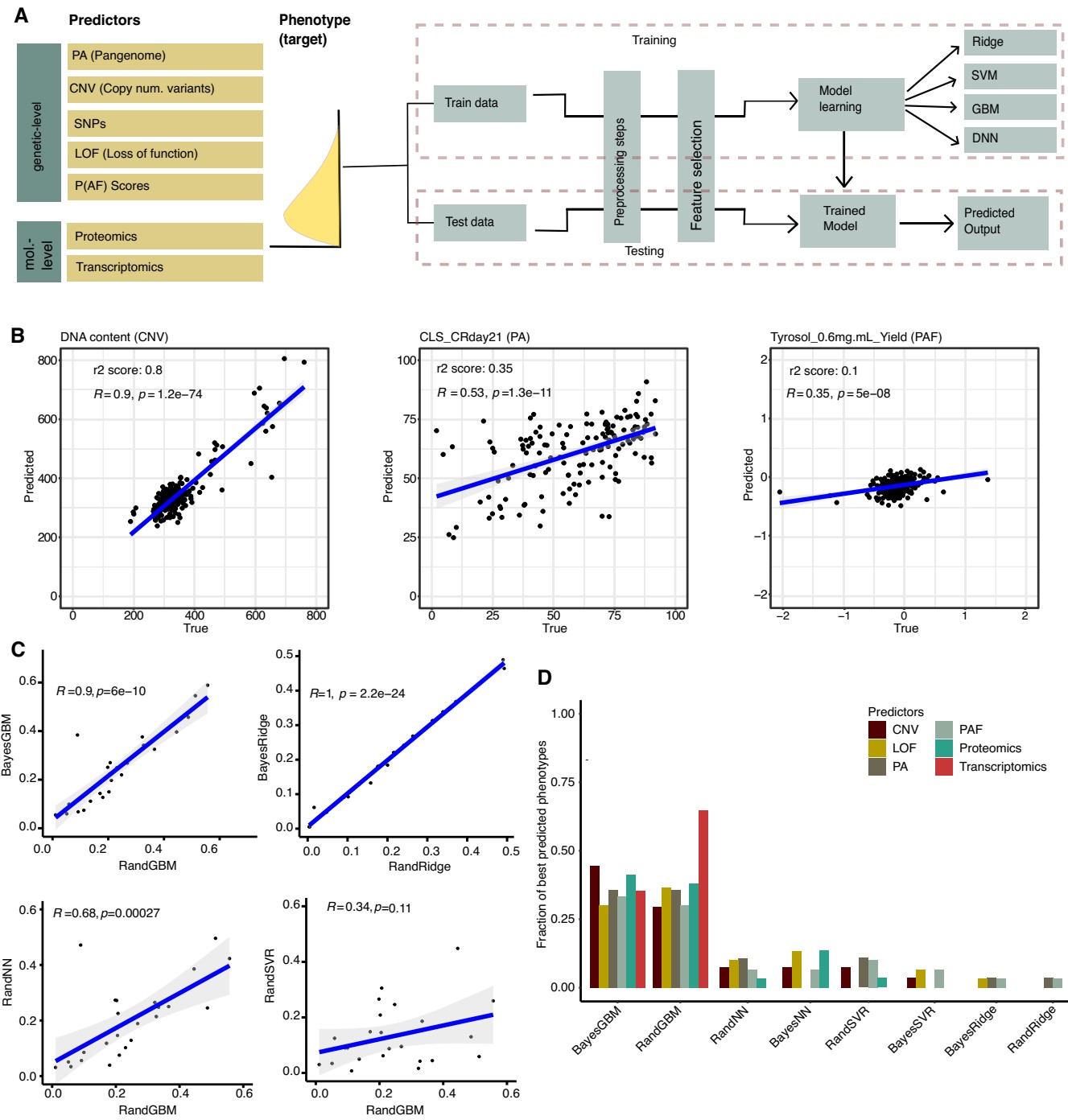

**Figure 3. Benchmarking of the Gen-phen pipeline.**

(**A**) Overview of the Gen-phen architecture. (**B**) Examples of good (DNA content), moderate (CLS CR day21), and poor (Tyrosol yield) phenotypic predictions (Pearson's correlation between true and predicted values) using BayesGBM. (**C**) Gen-phen benchmarking on 30 phenotypes using P(AF) scores as predictor shows high Pearson's correlation among the predictions from different methods except SVR. (**D**) Fraction of the best predicted phenotypes, calculated as the number of phenotypes that are best predicted by a specific method compared to the rest of the methods out of all 30 tested for each ML model across predictors. Bayesian GBM was the best method, followed by Random GBM (average 44% and 39% best predictions across different predictors, respectively).

("Methods" and Fig. 3A). The pre-processing step removes strains with missing phenotype values and randomly splits the dataset into a training (75%) and a testing (25%) set. For feature selection, the pipeline uses Lasso regression to select the features that best capture variation between strains and construct a sparse model that is computationally more efficient and generalizable for prediction.

In the model learning step, the user can choose between common ML algorithms (ridge, gradient-boosted machines

(GBMs), support vector regressor (SVR), and neural networks (NNs)) in combination with two types of optimization techniques (random and Bayesian) and decide the $k$ parameter of the $k$-fold cross-validation. While ridge regression is a linear regression method which uses L2 regularization to prevent overfitting, SVR utilizes the kernel trick to model complex, non-linear relationships. Furthermore, GBM is an ensemble of tree-based method that learns in a sequential manner using the residual errors from previous trees to build the next one. Finally, deep neural networks use multiple layered neural networks that can capture highly non-linear and complex patterns but require very large datasets. Model performance is evaluated using $r^2$ scores on the test set, quantifying how much variance in the target is explained by the model (Fig. 3B). The importance of individual features, which quantify their contribution to the predictions can also be extracted (weight coefficients for linear regression and Gini importance for GBM), except for SVR which use non-linear kernels and neural networks ("Methods").

Comparing the four feature selection strategies, we found LASSO grid selection to be the most efficient, with accuracy comparable to random LASSO selection and when no feature selection was performed across most models (Appendix Table S1; Appendix Fig. S6A). Using LASSO grid selection reduces the number of features considerably, thus decreasing computational times by several orders for some ML models (Appendix Fig. S6B). Hence, the eight model permutation strategies in Gen-phen were benchmarked more in-depth using 30 representative phenotypes from the eight phenotype classes with LASSO grid as feature selection criterion (Figs. 3C and EV2). For all four models, no significant difference was observed between the two hyperparameters optimization strategies (Fig. EV2A,B). A significantly high correlation was also observed between predictions from GBM and neural networks, which could be due to the non-linearity of the two methods, while SVR models showed the most deviation compared to other model predictions.

Overall, GBM Bayesian and Random regression models performed the best, giving the best predictions for ~75% of the 30 test phenotypes, followed by neural networks (16%) and SVR models (10%) (Fig. 3D). This suggests that non-linear ML models such as GBM and neural networks portray a better genotype–phenotype relation compared to linear models, such as ridge regression. Across all model types with 30 test phenotypes, pangenome gene presence-absence (PA) and P(AF) scores perform better, being superior predictors for an average (over model types) of 26% phenotypes individually. LOF (22%), CNV (14%) and proteomic data (5%) were less often the best predictors, while transcriptomics never ranked first (Fig. EV2C).

## Predictions across genomic and phenomic predictors

To determine which type of molecular data best predicts higher-level *S. cerevisiae* phenotypes, we compared different predictors as data inputs for the entire phenome using the best-performing BayesGBM model with no pre-feature selection. No significant difference between the distribution of accuracies among most predictors was observed with BayesGBM, except for transcriptomic that consistently were worse than other predictors (Fig. 4A). P(AF) scores and LOF exhibited highest similarity between their predictions (Pearson's coefficient, 0.86), which is expected as they are both based on deleterious mutants (Fig. 4B). Based on 223 phenotypes and mean cross-validation scores, we observed that P(AF) scores was the top predictor with best predictions in ~25% phenotypes followed by PA (~21%), implying genomic-level predictors to be more useful compared to molecular-level predictors such as proteomics and transcriptomics (Fig. 4C). However, no significant difference was observed between the predictions based on PA and P(AF) scores in addition to the two being highly correlated (Pearson's coefficient of 0.80), indicating most phenotypes are consistently predicted either well or poorly across predictors. When predicting higher-level phenotypes from the large SNP set (500 K variants), training time increased drastically (from 5 to 10 min. to >2 days per phenotype) and predictions were restricted to ~20 random phenotypes (Appendix Fig. S8A). BayesGBM did however not predict phenotypes from SNP data better than from other data types, hinting that the inclusion of genome-wide SNP datasets may not be necessary in ML phenotype predictions when other genomic data are available.

Next, we selected the BayesGBM (as best model) and the P(AF) scores (as best predictor), to explore how well different phenotype classes were predicted. Predicting all 223 phenotypes, we found large variations in model performance, both between and within phenotypic classes (Fig. 4D). Overall, the best-predicted phenotype class was sporulation, followed by growth in stress conditions (colony-size 2, median $r^2$ score = 0.38) and CLS, while growth under carbon and nitrogen source limitation was less well predicted. Traits predictability seems to be only weakly related to the intra-strain variability in phenotype measurements (Appendix Fig. S7A). Although no clear relation could be observed between coefficient of variation (COV) and the predictions (Appendix Fig. S7B), some trait types were significantly better predicted compared to others, e.g., growth yield compared to growth rate in the same conditions (Appendix Fig. S7C,D).

We also used BayesGBM to investigate to what extent a single higher-level phenotype can be predicted from a compendium of other higher-level phenotypes, e.g., through effects of pleiotropic genes or genetic linkage across genome regions, but without explicit genomic predictor information. We tested three different scenarios, basing predictions of single higher-level traits on all 223 such phenotypes, on only correlated phenotypes ($R > 0.3$ or $R < -0.3$) and on only uncorrelated phenotypes ($-0.3 < R < 0.3$), respectively. Overall, using correlated phenotypes resulted in excellent predictions, with very high accuracy ($r^2$ score >0.75) for 98 phenotypes and 36 showing near-perfect predictions ($r^2$ score > 0.90, Fig. 4E). The most predictive features are typically phenotypic measurements obtained under the same conditions but at different concentrations or time points (Fig. EV3A). Furthermore, we observed significant information sharing among similar nitrogen and carbon sources, leading to improved predictive performance in these conditions (Fig. EV3A,B). Similar to the phenotypic correlation map (Fig. 1B), we see that the same trait type (either yield or growth rate) measured in similar conditions is more useful for predictions than growth rate and yield measured in the identical condition (Fig. EV3B). Consistently, 13 phenotypes all disconnected from other phenotypes (Fig. 1B) were not well predicted by other phenotypes ($r^2$ score < 0.3) and including only uncorrelated phenotypes significantly reduces the predictions (Fig. 4E).

Finally, we explored whether adding metadata, in the form of strain ecological and geographic origin, heterozygosity level, phylogenetic clade and ploidy (1n, 2n, 4n) improved the accuracy

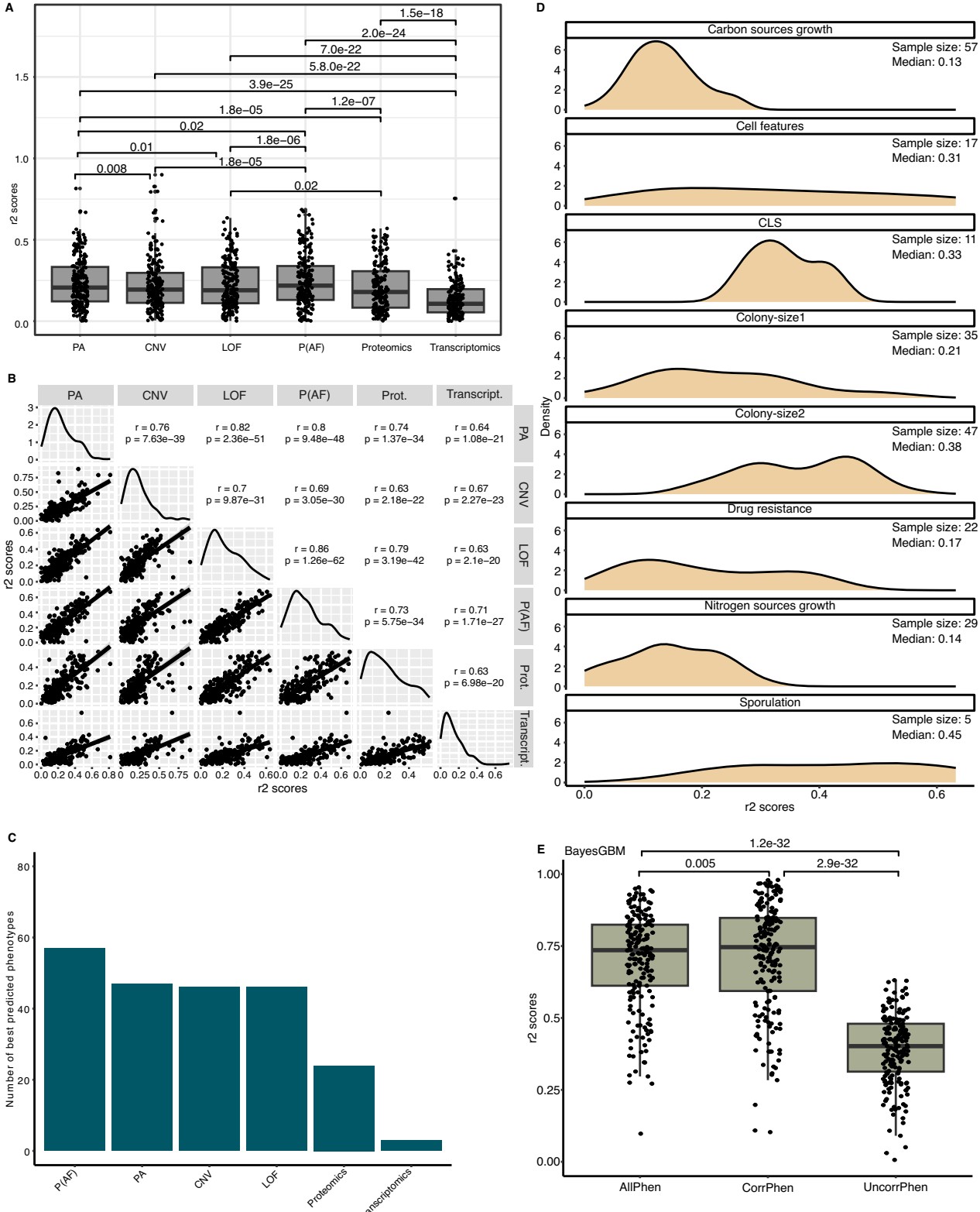

◄ **Figure 4.  Phenotype prediction accuracies.**

(A) Comparisons between the prediction accuracies across different genomic and gene expression predictors for the entire phenome ($n = 223$, Paired-Wilcoxon test). (B) Pairwise Pearson's correlation between the prediction accuracies shows most phenotypes to be consistently predicted well or poorly across the predictors. (C) Number of phenotypes best predicted by each predictor of the 223 phenotypes using BayesGBM. (D) Comparing prediction accuracies for all phenotypes partitioned by classes shows large variation. (E) Predicting phenotypes using the entire phenome (AllPhen), only correlated phenotypes (CorrPhen, with $R >= 0.3$ or $R <= -0.3$) and only uncorrelated phenotypes (UncorrPhen, $-0.3 < = R <= 0.3$) as predictors ($n = 223$, Paired-Wilcoxon test). The boxplots in (A, E) display the data as follows. The median (50th percentile) of the data is indicated by the black horizontal line, the lower and upper bounds of the box indicate the 25th (Q1) and 75th (Q3) percentile, respectively. The whiskers (vertical lines) extend to the smallest and largest value within 1.5* interquartile range (Q3-Q1), and the points beyond the ends of the line are potential outliers.

of BayesGBM predictions from different data types, but found no improvements (Appendix Fig. S7B). However, combining transcriptomics and proteomics data could improve the predictions compared to when these predictors were used individually (Appendix Fig. S7B). Similarly, a small improvement was observed when the top predictors were combined (Appendix Fig. S7B).

## Biological interpretation of machine-learning predictions

Next, to understand which individual features influenced predictions the most, and why, we systematically extracted the feature importance (FI) scores (Huynh-Thu et al, 2012) ("Methods") across all phenotypes and inspected whether the highest-scored features were enriched in relevant gene candidates and biological processes. Most variants exhibit low FI scores with only few standing out, and their strength was mostly independent of the variant frequencies in the population (Fig. 5A,B). Several SNPs with high FI scores were present at low frequencies, consistent with previous reports of outsized effects of rare variants on yeast phenotypic variation, and most of these were predicted loss-of-function variants (Bergström et al, 2014; Bloom et al, 2019) (Fig. 5A). We observed few high FI scores for phenotypes corresponding to growth measured in $CuSO_4$, dyads production and in NaCl, retrieving as top candidates in these respective environments, copy number for YHR055 C (*CUP1*), LOF in YHR152W (*SPO12*), and transcriptome for YDR040C (*ENA1*) and YDR039C (*ENA2*) respectively (Figs. 5B and EV4A). The gene *SPO12* LOFs had previously been connected to trigger dyads formation (De Chiara et al, 2022), while the copy number variation of the *CUP1* and ENA genes are well known to be associated with yeast growth in $CuSO_4$ (Peter et al, 2018) and NaCl (Warringer et al, 2011) environments respectively (Fig. 5C). The LOF in *SPO12* is rare (present in <0.01 of the population) and therefore undetectable through GWAS (Fig. 5C). This suggests that ML can identify at least some rare genetic features of high impact associated with the phenotypes. For the case of NaCl, the *ENA1* and *ENA2* transcript levels are top FI scores across all concentrations and time points when transcriptomics was used as a predictor (Fig. EV4A), whereas when CNV was used as predictor, population structure seems to be as informative as the *ENA* copy number (Fig. EV4B). This is consistent with the presence of two major alleles of *ENA*, a reference-type that includes *ENA1, 2, 5, 6,* and a highly diverged *ENA*-like (Peter et al, 2018; Warringer et al, 2011), with their distribution closely following the population structure (Fig. EV4B).

The number of features selected and the distribution of their scores varied significantly depending on the phenotypes. Analysis of top 50 FI scores across phenotypes depicts a low FI score average

(less than 0.01) for most predictors, suggesting limited individual feature-level contribution for most cases (Appendix Fig. S9). Additionally, phenotypes that have the highest average FI scores vary across the predictor types. For the cases where several genes had moderate FI scores, we performed functional enrichment to investigate the underlying biological processes (Appendix Fig. S10). Genes with high FI scores (>0.01) for growth in sucrose (based on P(AF) scores) were enriched in protein binding, hydrolase activity, ligase activity, whereas ketoconazole (based on SNPs) were enriched in regulation of biological and cellular processes (Appendix Fig. S10).

## Discussion

We exploited the *S. cerevisiae* 1011 framework to investigate the genotype–phenotype landscapes through ML. The availability of hundreds of phenotypes measured under controlled environments enabled us to explore phenotypic correlations at scale. We observed a strong impact of trait-type over environmental conditions in which the traits were recorded. Unexpected correlations between unrelated phenotypes also emerged, with several instances that were independently recapitulated using the gene-deletion collection (Turco et al, 2023) demonstrating their biological basis, although only a modest overlap is expected between phenotypic consequences of natural variation and artificial deletions (Jiang and Zhang, 2023). One interesting example is the correlation between the apparently unrelated CLS and azole resistance phenotypes, emerging from both the natural and the deletion collections. While the basis of this overlap is unknown, possible mechanism candidates include sterol lipids' role in both membrane structures and signaling and genes involved in cell wall integrity (Matecic et al, 2010; Garay et al, 2014; Anderson et al, 2003).

The genetic signals underlying phenotypic correlations can be captured by ML and thus improve predictions. Part of these signals are driven by pleiotropic variants that were also detected by GWAS, but some are driven by genetic determinants that escaped GWAS detection, such as epistatic interactions and rare variants. One unexpected finding of our study was the ability of ML to identify rare variants with high impact, and their causative effects were supported by experimental validation. This points to ML being able to at least partially overcome one of the major limitations of GWAS. Rare variants, and rare epistatic interactions, are nevertheless likely to limit the power also of ML predictions in natural populations from individual genomes. Controlled crosses, where variants and variant pairs that are rare in natural populations are at much higher frequencies, can allow a more complete

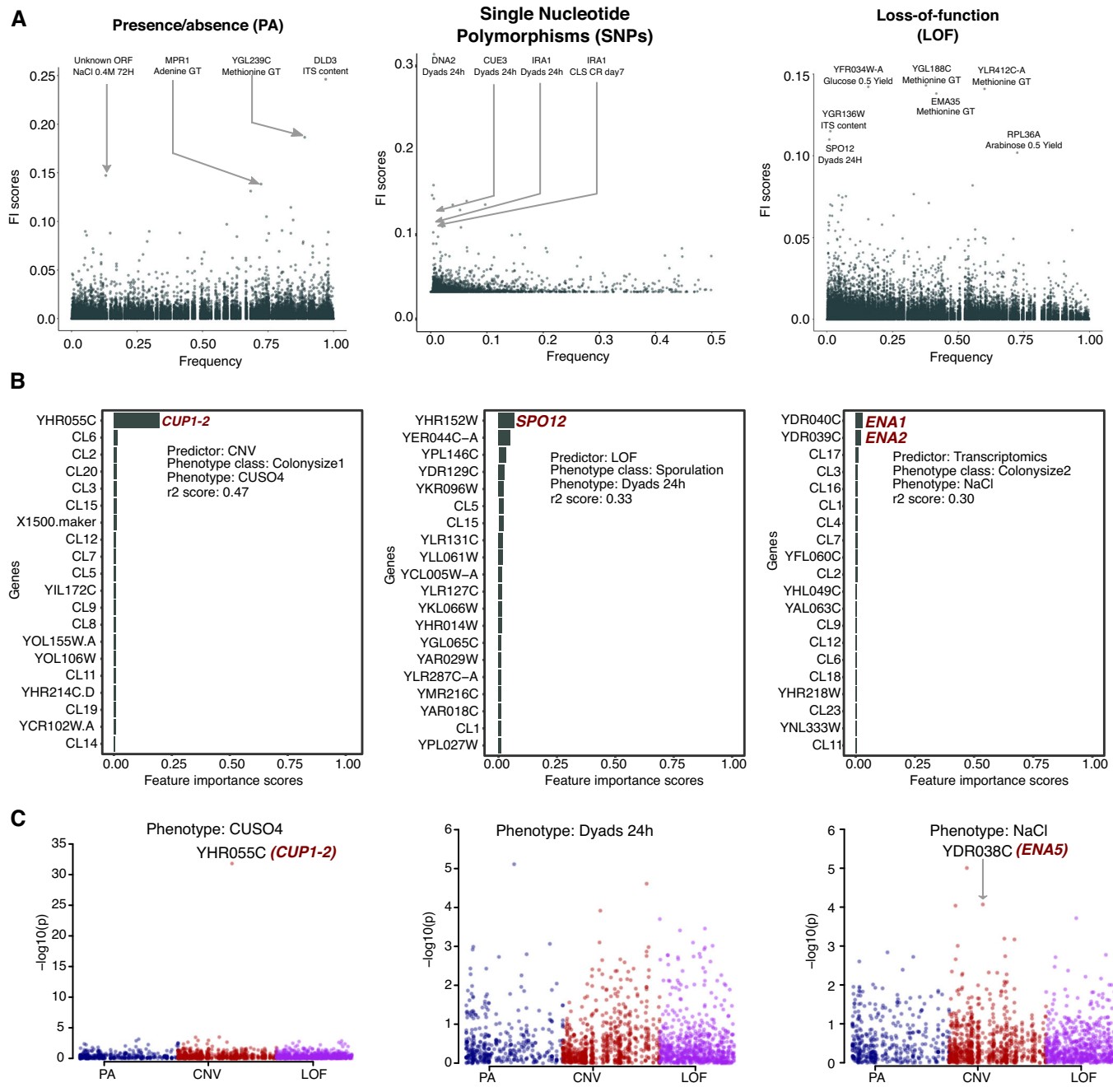

**Figure 5. Portrait of the predictors FI scores.**

(A) The strength of the FI scores for PA and LOF predictors is independent of their frequency in the population, while SNPs with the highest FI scores are enriched for rare variants. The highest-scoring variants are annotated. (B) Top-ranked feature importance score genes for growth in CuSO$_4$, dyads production, and NaCl (averaged over 10 iterations). (C) The corresponding GWAS output for the three phenotypes showing the *CUP1-2* and *ENA5* gene appearing as hits in CuSO$_4$ and NaCl growth phenotypes, while the *SPO12* is too rare to be detected by GWAS.

decomposition (Hallin et al, 2016) and accurate prediction (Märtens et al, 2016) of such traits. GWAS hits also included equal proportions of variants deriving from synonymous and missense mutations, suggesting a non-neutral impact of synonymous mutations on complex phenotypes. Although the high frequency of synonymous mutation detected by GWAS could partially be derived from linkage disequilibrium, which limits the

detection of the exact causal variants, their impact on complex traits remains an open question (Hunt et al, 2014; Bailey et al, 2021; Shen et al, 2022; She and Jarosz, 2018). While several causative variants were previously validated, other SNPs including synonymous mutations, did not show detectable phenotypic effects when reconstructed in different backgrounds. This might be due to multiple factors, including our ability to detect small effects on

phenotype from low-impact variants and the genetic background dependency. The effect of yeast genetic background has been found to be strong also on gene deletion phenotypes (Galardini et al, 2019).

The construction of the Gen-Phen ML pipeline enabled us to systematically compare ML models and input predictors. Non-linear models outperformed the linear ones, with gradient boosting trees emerging as best-performing model, suggesting a predominantly non-linear relationship between the genotypes and phenotypes. Unexpectedly, transcriptomics was a poor predictor for most phenotypes, specifically compared to other molecular-level data such as proteomics. Although we lack a clear understanding for this, studies have shown stronger control of the protein regulation through post-transcriptional buffering compared to mRNA expression which is impacted by transcriptional noise (Teyssonnière et al, 2024).

Recent studies on a macroevolutionary time-scale have used over 1000 Saccharomycotina yeasts to train ML models to predict discrete growth phenotypes and ecological lifestyles from various genomic, metabolic and phenomic data (Opulente et al, 2024; Harrison et al, 2024; Gonçalves et al, 2024). Such approaches were also highly successful in predicting bacteria categorical antibiotic resistance (Moradigaravand et al, 2018), and moderately successful in predicting bacterial growth at sub-inhibitory concencentrations (Benkwitz-Bedford et al, 2021). Our work complements these studies at the microevolutionary scale by exploring prediction of quantitative phenotypes within a single yeast species. While a direct comparison of prediction performance among these studies is challenging, pangenome variation in the form of gene presence and absence emerge as strong predictors at both micro- and macro-evolutionary timescales.

A limitation of our study is the number of samples available, and increasing the sample size will certainly improve both the accuracy and the generalizability of the models. Combining different prediction features did however not improve prediction accuracy. This might be due to the small sample size and the limitations of the ML methods used to model the complex interactions between different types of data and suggests the need for larger datasets along with multimodal ML methods that enable the integration of heterogeneous data (Argelaguet et al, 2018; Tan et al, 2021; Parisot et al, 2018; Nguyen et al, 2022). Prediction accuracy among the test strains varies depending on their genetic distances from the training strains, with strains that are closely related better predicted than distantly related ones. This suggests that our framework may have better prediction accuracy in higher eukaryotes with less genetic diversity, such as humans. Moreover, phenotypes measured in stress environments were predicted better than the phenotypes measured in nutrient-restricted environments, such as different types of carbon and nitrogen limitations. This was not unexpected since growth restricted by the availability of different carbon and nitrogen sources are high-complexity traits controlled by many enzymatic and regulatory genes and variants. Stress conditions are often simpler, with variants in a few genes responsible for excluding the stressor outside from cells having outsized phenotypic impact and being easier to detect (Hallin et al, 2016; Vázquez-García et al, 2017). While trait complexity is likely contributing to the prediction performance, identifying other contributing factors is challenging.

Overall, our study provides a framework to quantify ML phenotypic predictions and to identify variants likely impacting phenotypes. This included rare variants in *SPO12* gene previously linked to dyads formation (De Chiara et al, 2022) and in the negative regulator *IRA1* of the RAS /cAMP signaling pathway a known target leading to natural phenotypic variation (Parts et al, 2011; Aguilar-Rodríguez et al, 2024). While prediction accuracy varies greatly across phenotypes, larger datasets can be further used to explore whether prediction accuracy improves and might enable us to identify common variants with small effects. Beyond the results presented, we provide a flexible ML workflow that can be easily adapted to different species and data types, including in clinical settings as an efficient way to predict phenotypes.

# Methods

**Reagents and tools table**

| Reagent/resource | Reference or source | Identifier or catalog number |
|---|---|---|
| **Experimental models** | | |
| *S. cerevisiae* | https://doi.org/10.1038/s41586-018-0030-5 | http://1002genomes.u-strasbg.fr/files/) |
| **Recombinant DNA** | | |
| **Antibodies** | | |
| **Oligonucleotides and other sequence-based reagents** | | |
| CRISPR gRNA: SEC11_gRNA_Cut1 | This study | DATASET_EV5 |
| CRISPR gRNA: SEC11_gRNA_Cut2 | This study | DATASET_EV5 |
| SEC11_Repair_FW | This study | DATASET_EV5 |
| SEC11_Repair_RV | This study | DATASET_EV5 |
| **Chemicals, Enzymes and other reagents** | | |
| Propidium iodide | Sigma-Aldrich | Cat # P4170-10MG |
| YO-PRO-1 iodide | Thermo Fisher | Cat # Y3603 |
| RNAse A | Sigma-Aldrich | Cat # R6513-10MG |
| Triton X-100 | Sigma-Aldrich | Cat # X100-5ML |
| Rapamycin | Sigma-Aldrich | Cat # R8781-200UL |
| **Software** | | |
| scikit-learn 1.3.2 library | https://doi.org/10.48550/arXiv.1201.0490 | |
| PyTorch library V1.13.1 | https://doi.org/10.48550/arXiv.1912.01703 | |
| Variant effect predictor | https://doi.org/10.1186/s13059-016-0974-4 | |
| fast-lmmc v2.07.20140723 | https://doi.org/10.1038/nmeth.1681 | |
| PLINK v1.90b6.18 | https://doi.org/10.1086/519795 | |
| FlowJo™ v10 | BD Life Sciences | |
| **Other** | | |

## Phenotypes for the natural population

The 1011 collection represents a natural, non-genetically manipulated population of *S. cerevisiae* species, unlike deletion collections.

It has been phenotyped by multiple laboratories in different studies. We included 190 published phenotypes from four main publications (De Chiara et al, 2022; Peter et al, 2018; D'Angiolo et al, 2023; Galardini et al, 2019) and 33 unpublished phenotypes of which 22 phenotypes belong to growth measured in eight different antifungal drugs (https://github.com/SakshiKhaiwal/CEFIPRA) and 11 independently measured phenotypes, including cell size, DNA content, cell flocculation frequency, and CLS measured in rapamycin which are released in this study. Cell size and flocculation were estimated by flow cytometry. Each strain was grown in YPD media to the exponential phase in 96-well microtiter plates prior to fixation with ethanol 70%. Cells were washed twice with PBS and resuspended in a DNA staining solution (15 µM propidium iodide (Sigma-Aldrich), 100 µg/ml RNAse A (Sigma-Aldrich), 0.1% Triton-X, in PBS solution) and incubated for 3 h at 37 °C in the dark. A total of 10,000 cells were directly analyzed on a FACS Calibur flow cytometer using the FL2-A channel. The red fluorescence generated by stained DNA was used to identify isolated unbudded (1N) and budding (2N) cells, from which DNA content was estimated. Instead, cell size was inferred based on the FSC-H value of either unbudded and/or budding cells. The frequency of isolated 1N and 2N cells out of the total events was then leveraged to determine the extent of flocculation. Four replicates of the lab strain BY4743 were appended into each microtiter plate for data normalization. The determination of CLS in rapamycin was performed in synthetic media supplemented with 0.025 µg/ml of drug as described here (Barré et al, 2020). The phenotypes corresponding to doubling time have been inversed to represent the growth rates, so a higher value would mean faster growth and vice versa. The phenome is enriched for the yield trait obtained from three main datasets (Peter et al, 2018; De Chiara et al, 2022; Galardini et al, 2019). In all the cases, yield is inferred from the size of the yeast colonies measured at a given time point. The entire set of phenotypes, along with additional information corresponding to strains such as clades, ecological and geographical sources of isolation is given in Dataset EV1. In addition, the information regarding the replicates and their further analysis is provided in Dataset EV2.

We divided the phenotypes into eight major classes to make comparisons between various groups of environments, traits, and the experimental strategy (Appendix Fig. S1A). For example, CLS, sporulation, and cellular traits are distinguished from growth traits (yield/colony-size and growth rate). Growth rate and yield phenotypes which constitute 94 phenotypes measured using scan-o-matic (Zackrisson et al, 2016) in three main types of environments, i.e., carbon-rich conditions, nitrogen-rich conditions, and antifungal drugs are separated into three classes, namely "carbon sources growth", "nitrogen sources growth", and "drug resistance", respectively. This was done to quantify the impact of the environment on the phenotypes. The intra-strain replicate variance for these types of experimental measurements has been previously studied in depth (Zackrisson et al, 2016; Borse et al, 2024).

Finally, colony sizes measured in similar stress conditions such as temperature, chemicals, and drugs constituting 35 and 47 phenotypes, respectively, are obtained from two studies (Peter et al, 2018; Galardini et al, 2019) from different labs that allow us to compare the batch effects, experimental techniques, and timepoints for similar traits and environments. The phenotypes with defined classes with related publications are provided in Dataset EV1.

## Gene-deletion collection phenotypes

The *S. cerevisiae* gene-deletion collection refers to the collection of strains with each strain having a specific non-essential gene deleted systematically and consists of 4889 strains (Winzeler et al, 1999). We extracted phenotypes from the gene deletion collection (yeastpheno-me.org), measured for similar traits and conditions of the ones available in the 1011 *S. cerevisiae* collection. We manually filtered the gene deletion collection first by traits (e.g., CLS, colony-size, sporulation) and then using keywords for environmental conditions. We obtained a list of 200 overlapping phenotypes (Dataset EV3).

## Genomic predictors

Genomic matrices which include the matrix of presence/absence (PA) and copy number variations (CNV) of ORFs, and SNPs are obtained from (Peter et al, 2018). The pangenome of 1011 collection used consists of 7796 ORFs, out of which 4940 are present in the core genome, while 2856 that are variable within the population. The PA matrix consists of whether a given ORF is present or absent depicted by '1' and '0' respectively, in a given strain. The CNV matrix consists of the number of copies for each ORF per strain for the haploid genome, with no genomic position information. The loss of function matrix includes all nonsense and frameshift mutations with '1' and '0' depicting the presence and absence of loss of function in a gene, respectively (Dataset EV9). Finally, the SNPs matrix represents all SNPs filtered with a minimum allele frequency (MAF) at 0.5%. The table consists of combinations of '0', '1', '2', and '3'. The homozygous reference is '0'. Other possible combinations are 1.1, 2.2 and 3.3 for homozygous alternative alleles or 0.1, 0.2, 0.3, 1.2, 1.3 and 2.3 for all possible heterozygous combinations.

## Population structure

The population structure is calculated using the same strategy as presented in (Moradigaravand et al, 2018). The pairwise SNPs distance matrix was obtained from (Peter et al, 2018) and the clustering is performed using the gengraph function of the adegenet package in R with the threshold range defined from 0.001 to the maximum value in the SNP distance matrix that varies the step size of 0.01 which gives us the range of clusters from 1 to 103.

## P(AF) scores

The P(AF) scores (Jelier et al, 2011) that combine the functional effect of all variants in a gene were calculated for 1011 strains using the VariantAnnotation R package (Obenchain et al, 2014; Dunham, 2021) (Dataset EV9).

## Proteomics and transcriptomics

The proteomics and transcriptomics data are obtained from (Muenzner et al, 2024) and (Caudal et al, 2024), respectively.

## Phenotypic analysis

The correlation map is constructed using the corrplot 0.92 library in R 4.3.0. Phenotype networks are built using the correlation

matrix using the network and ggnet2 functions of the network and ggplot2 library in R. We separated correlation and anticorrelation in distinct networks and filtered out weak correlations/anticorrelation using 0.5 and −0.3 thresholds, respectively (Fig. 1D; Appendix Fig. S1A). The phenomic distance between each pair of strains was calculated as the sum of the absolute differences of all the phenotypes, and the tree was constructed using the biojn function from the ape library v5.7-1.

## GWAS

We mined GWAS results from 99 phenotypes previously reported (De Chiara et al, 2022). We re-run GWAS for 81 phenotypes using the same genetic data input and method to ensure consistency. Furthermore, we performed GWAS on 42 novel phenotypes that have not been included in any previous studies. GWAS are performed using similar settings and genome matrices (SNP, CNV, PA, and LOF) (Peter et al, 2018). Genotype markers with <5% MAF are removed from the analysis. Linear regression was implemented using fast linear mixed models using fast-lmm (fast-lmmc v2.07.20140723) (Lippert et al, 2011). The phenotypes are scaled to a standard normal distribution using the qqnorm function in R. We used the 5% family-wise error rate as a threshold to extract significant markers related to a trait. This threshold for the $p$-value is obtained by performing 100 permutations of the phenotypes and extracting the fifth lowest $p$-value. The significant GWAS hits obtained from the GWAS analysis are annotated, and for the SNPs that are not present in genes, we reported the flanking genes.

## GWAS error rates and limitations

Classical methods, such as the Bonferroni correction, to control the family-wise error rate (FWER), have proven to be overly conservative as they ignore that many variants co-vary due to linkage in the genome and do not associate independently with traits (Kaler and Purcell, 2019; Perneger, 1998). Performing empirical simulations to estimate the FWER for every trait individually is a widely accepted and standard practice in the GWAS field (Peter et al, 2018; De Chiara et al, 2022; Caudal et al, 2024; Togninalli et al, 2018). Estimating a significance threshold for each trait individually has the added benefit of accounting for the often substantial variations in genetic architecture across phenotypes (Kaler and Purcell, 2019). Here, we control the FWER per-trait at 5%, i.e., we set the significance threshold such that there is a 5% chance of having at least one false positive per phenotype. While we acknowledge that some false positive will be present in our yeast GWAS catalog, this threshold provides a good balance between false positives and negatives.

An alternative way to control the type I error is to use a false discovery rate (FDR) correction. A conventional method to perform FDR correction is using the Benjamin–Hochberg method, which is less conservative than the Bonferroni correction but still has the limitation due that it assumes independence among the tested hypotheses (Benjamini and Hochberg, 1995; Kaler and Purcell, 2019). This is not the case for our data, both because of the genetic co-variation between variants due to linkage, and because many variants are pleiotropic and thus have co-varying effects on phenotypes (Fig. 1C). We thus expect it to be highly conservative and result in a high rate of false negatives. For comparative

purposes, we nevertheless also explored results using the FDR correction. We combined all the SNPs, CNVs, PA and LoF in one genotype matrix of ~100k genotypes, tested these individually against the 223 phenotypes for a total of 20 million tests. We then assumed all these tests to be independent, and used the Benjamini–Hochberg procedure to obtain the adjusted $p$-values (or $q$-values). Using a threshold of $q = 0.05$, we obtained 887 unique GWAS hits (1662 less compared to the previous analysis) that were significant at this significance level. Out of the 887 GWAS hits, 709 hits overlapped with the GWAS hits from the GWAS hits called using the FWER correction. Among the GWAS hits only detected by the FWER approach were many high-confidence calls, including the experimentally validated SNP in AIM29. This likely reflects the expected higher rate of false negatives when performing an FDR correction. To provide a more complete GWAS hits catalog that captures more true biology, and be able to better compare GWAS hits to ML predictions, we use GWAS hits called by the FWER-based correction for the majority of our analyses. We encourage downstream users to perform additional fine-mapping or downstream experimental analysis to confirm the causal variants. We further predicted the effect of the variants using a Variant Effect Predictor (VEP) (McLaren et al, 2016) and categorized them according to their consequences, such as synonymous, missense, and loss of function. The general results do not change, and there are no significant differences between the adjusted $p$-values distribution among different categories (Appendix Fig. S11).

## Linkage disequilibrium (LD) filtering

To remove bias due to linkage disequilibrium (LD), we calculated LD scores for all pairs of SNPs that were within a 1 kb window and less than 20 SNPs apart from each other. For pairs of SNPs with an LD score greater than 0.5 and significantly associated with the same phenotype, we only consider the SNP with a higher significance value (or lower $p$-value). LD scores are calculated using PLINK v1.90b6.18 (Purcell et al, 2007).

## Variant effect predictions

The consequences of all the variants including the variants found to be significantly associated with specific phenotypes are calculated using the online web tool called Variant Effect Predictor (VEP) (McLaren et al, 2016) with a buffer size of 250 and upstream/downstream distance of 300 bp. To predict the impact of the missense variants to identify deleterious mutations, we used the mutfunc database and extracted all the predictions corresponding to the SNPs present in the pangenome of the *S. cerevisiae* (Wagih et al, 2018).

## Network expansion using the Personalized Page Rank (PPR) algorithm

Network expansion analysis is performed as described in (Barrio-Hernandez et al, 2023). To construct the interactome of the proteins, two sources of interactions namely STRING and BioGRID were used. To expand the signal, we used BioGRID and STRING with a combined score >=0.4. We used 173 phenotypes with at least 2 genes associated with an SNP (the bare minimum to start the network expansion). We ran the method and used the personalized

pagerank scores to measure the distance between the different traits in the study. To identify traits controlled by the same biological pathways, we extracted genetic clusters with more than 75% overlapping genes between different traits.

## Gen-phen prediction pipeline

The Gen-phen pipeline is built in Python 3.8 using the models from the scikit-learn 1.3.2 library (Pedregosa et al). The pipeline comprises four major steps: pre-processing, feature selection, model learning, and model testing.

### Preprocessing

The Preprocessing step includes steps such as removing noisy data, imputing missing data, scaling the data, and splitting the dataset into a training and test set. The type of imputation strategy (mean or k nearest neighbors imputation) can be selected by the user, in addition to the fraction of data to be used as test set (default = 0.25). We have implemented three types of splitting criteria to split data into a training and a testing set, namely Hold-out at random (HOAR), Intra-clade hold-out (INHO), and Leave-one-clade-out (LOCO). HOAR involves randomly dividing genotypic and phenotypic data for all strains into a 75% training set and a 25% testing set, while INHO strategy only considers the strains belonging to the Wine European strains (~300) and randomly assigns 50 of them to the test set and the rest to the training set. Finally, the LOCO strategy utilizes one entire clade as a test set, which in our case is 'M3.Mosaic_Region_3', and uses the rest of the clades for training. The data was scaled using standardscaler() function from the sci-kit learn library that standardizes the data on a normal scale so that the data has a mean 0 and variance 1. The missing data in the features was imputed with the mean, while the samples missing the phenotypic data were removed from our analysis.

### Feature selection

LASSO regression (Ranstam and Cook, 2018) is a commonly used machine-learning technique that adds a penalty term to the regression function and sets the coefficient of some of the features to be zero which can be discarded from the data. The feature selection is implemented in two ways: 1) using the LASSO regression from scikit-learn and the high-dimensional LASSO regression using the Hi-LASSO Python library 1.0.6 (Jo et al, 2022). Lasso selection is implemented through grid, random, and Bayesian search hyperparameter optimization. The grid and random hyperparameters optimization are done using GridSearchCV and RandomizedSearchCV function from scikit-learn while Bayesian hyperparameters optimization is done using BayesSearchCV function from scikit optimize 0.9.0 (Head et al, 2018). The grid selection is run with fivefold cross-validation and with alphas equal to 0.001, 0.01, 0.1, and 0.5 for 10,000 iterations. Random and Bayesian optimization is carried out using fivefold cross-validation with alpha selected from a log-uniform distribution with values ranging from 1e-4 to 1 up to 500 and 200 iterations, respectively. The hi-lasso function is used with q1 and q2 being set to "auto" with L and alpha being 30 and 0.01, respectively.

### Model learning

The model learning is implemented using 5 ML methods, namely ridge regression, elastic regression, support vector regression,

gradient-boosted trees, and deep neural networks in combination with two types of hyperparameter optimization techniques, namely random and Bayesian search optimization along with fivefold cross-validation. All the hyperparameter distributions used for each model are defined in the "model" part of the Gen-phen pipeline. Bayesian optimization is used with 100 iterations, whereas random optimization is used with 1000 iterations, except for the case of neural networks where we use 100 iterations due to their high computational power consumption. The number of folds for cross-validation and the number of iterations can be chosen manually by the user.

Ridge regression.  Ridge regression (Hoerl and Kennard, 1970) is implemented using the Ridge function from the class linear models of the scikit-learn library and requires optimization of the hyperparameter, α which is a constant that controls the regularization of the L2 norm. For randomized optimization, we used alpha from a log-uniform distribution with values ranging from 1 to 1000 while for Bayesian ridge regression, alpha is taken from a uniform distribution ranging from 1 to 10,000.

Elastic net regression.  Elastic net regression (Zou and Hastie, 2005) is also implemented using class linear models through the ElasticNet function from scikit-learn. It uses a combination of L1 and L2 and therefore requires optimization of two hyperparameters, α which is the constant that controls the penalty terms, and the L1 ratio known as the mixing parameter that controls the ratio of L1 and L2 regularization. For both randomized and Bayesian optimization, we used uniform distributions with values ranging from 0.001 to 1 and from 0 to 1 for α and L1 ratio, respectively.

Gradient-boosted decision trees.  Gradient-boosted decision trees (Friedman, 2001) are ensemble-based methods and are implemented using the GradientBoostingRegressor function from the class ensemble in the scikit library. There are a number of hyperparameters that need to be defined for gradient-boosted trees. Some of them are the loss function, which is defined to be as "squared error" in this case, "learning rate" which is a parameter to control the contribution of each tree to the prediction, 'n estimators' to define the number of decision trees that are trained in the ensemble and so on.

HistGradientBoosting regressor.  This histogram binning-based gradient boosting regression (Ke G et al, 2020) method has an inherent support for missing values, and is particularly useful for datasets above 10 K. The hyperparameters to be optimized, include "learning_rate", "max_depth" for maximum depth for each regressor, "max_leaf_nodes" for maximum leaf for each tree, etc.

Support vector regression.  Support vector regression (Awad and Khanna, 2015) focus on trying to find a hyperplane that minimizes the error between the predicted and actual target values. This hyperplane could be a linear or non-linear function depending on the kernel used to map the data from a lower-dimensional to a higher-dimensional space for better clustering. Some of the hyperparameters to be optimized are 'kernel' for the type of kernel, the error margin ε is the range in which the loss function is equal to zero for data points that are predicted within it, regularization parameter 'C', and the kernel coefficient 'gamma'. It is to be noted

that the non-linear kernels in SVM transform the original feature space to a complex and non-linear higher-dimensional space, making the calculation of feature importance scores for those cases difficult.

Deep neural networks We have utilized the fully connected feed-forward neural networks for individual phenotype predictions (LeCun et al, 2015). The architecture of these types of deep neural networks is defined mainly by two parameters, the number of hidden layers and the number of neurons in each hidden layer. A custom function is defined to design the neural network architecture depending on the dimension of the input layer and the percentage decrease in the number of nodes in each successive layer. This gives a neural network with a constant decrease in the number of nodes in the hidden layers. Some other hyperparameters that are optimized while training include α for the strength of the regularization term, activation for hidden layers, solver for weight optimization, and batch size for mini-batches size for stochastic optimizers.

### Model testing

The accuracy of the model is assessed using the held-out test strains which consist of ~150 strains that have not been used for training. Several statistical quantities are reported to quantify prediction performance such as $r^2$ scores, correlation between the predicted and experimentally measured values measured using Pearson's coefficient, mean squared error, and mean cross-validation scores. Moreover, feature importance scores (Zien et al, 2009) (wherever valid) are also reported to quantify the importance of individual features to predictions.

### Feature importance calculation

For ridge, elasticnet and linear SVR models, feature importance scores are defined as the absolute weight coefficients associated with the features (Zien et al, 2009). The higher the absolute weight, the more important the feature is. In the case of gradient boosting trees, the feature importance is calculated based on impurity criterion. The importance of a feature is calculated as the reduction in the error criterion ("friedman_mse" or "squared_error") brought by that feature, known as Gini importance (Krzywinksi and Altman, 2017).

## Multivariate target predictions

We used three regression-based methods, multitask LASSO which is a linear-regression-based method capable of building sparse models, a tree-based method namely multi-regression gradient boosting decision trees, and a neural network-based method that can be used to build highly non-linear and complex methods. To implement multitask LASSO, we used the `linear_model.MultiTaskLasso` function from the scikit-learn library along with grid-search hyperparameters optimization with three-fold cross-validation. We used the "MultiOutputRegressor" from "sklearn.multioutput" along with the RandomizedSearchCV optimization to implement the multi-target gradient boosting decision trees regression. Finally, to construct a neural-network-based multitarget regression model, we used a U-Net architecture with residual connections (Ronneberger et al, 2015; He et al, 2016). This kind of architecture has the advantage of being easier to train, notably by avoiding the problem of vanishing gradients. This problem occurs particularly in deep neural networks, resulting in inefficient updating of the weights in the first layers (He et al, 2016). The

PyTorch library V1.13.1 (Paszke et al, 2019) with Python 3.8 was used for the construction of the neural net. The training, validation, and test sets were split randomly with 50%, 30%, and 20% strains, respectively, followed by missing values imputation using the mean of the data for both features and target. The number of neurons in the input and the output layers was set to be equal to the number of features and the number of phenotypes to be predicted, respectively. The model was built with eighteen hidden layers in addition to an input and output layer with three residual connections from the three layers at the beginning of the network to the three layers at the end of the network. We also used batch normalization which helps to increase the training efficiency and stability by recentering and rescaling the input layer. Furthermore, to regularize the training, dropout rates of 0.5, 0.25, and 0.25 were used for the beginning and end layers and middle layers, respectively. Finally, a learning rate of 0.025 was used with the mean squared loss function and Adam optimizer ($lr = 0.0000001$, weight_decay $= 0.1$). $r^2$ scores and Pearson's coefficient between the predicted and the true values were used as a measure of accuracy, and running accuracy is calculated as the average accuracy of the samples in the validation set. The training was performed for 200 epochs with validation being done once every five epochs. The corresponding results and figures are provided in Appendix Note S2 and Appendix Figs. S12–S15.

The details of all the parameters and hyperparameters used for the individual phenotype prediction pipeline and multi-phenotype predictions can be found in the scripts at: https://github.com/SakshiKhaiwal/Genotype-to-phenotype-mapping-in-yeast/tree/main.

## Statistics

We used the pairwise Wilcoxon test for two-group comparisons, which is a non-parametric test and does not assume data to be normally distributed, using the function wilcox.test in R v.4.3. T test was used for normally distributed replicate measures (Fig. EV2A). Multiple hypothesis testing were performed using FDR correction at α = 0.05.

## Data availability

All the phenotype data used in this study are available as appendix datasets as detailed in the text. The unpublished antifungal drug resistance data acquired within the CEFIPRA project "The genomic and evolutionary landscape of azole resistance in budding yeast" are available at: https://github.com/SakshiKhaiwal/CEFIPRA. The genomic predictors matrices used for GWAS and ML predictions are obtained from https://static-content.springer.com/esm/art%3A10.1038%2Fs41559-022-01671-9/MediaObjects/41559_2022_1671_MOESM5_ESM.gz. The transcriptomics dataset for the 1011 S. cerevisiae collection was obtained from http://1002genomes.u-strasbg.fr/files/RNAseq. The proteomics dataset for the 1011 S. cerevisiae collection was obtained from https://static-content.springer.com/esm/art%3A10.1038%2Fs41586-024-07442-9/MediaObjects/41586_2024_7442_MOESM3_ESM.xlsx. The developed computational pipeline and scripts are available at: https://github.com/SakshiKhaiwal/Genotype-to-phenotype-mapping-in-yeast/tree/main.

This software is provided solely for the purpose of scientific reproduction of the research described in the associated article and general academic research. Any other use, including commercial use, is prohibited without prior written consent from the authors. All rights reserved.

The source data of this paper are collected in the following database record: biostudies:S-SCDT-10_1038-S44320-025-00136-y.

## Peer review information

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

## Acknowledgements

We thank Alan Moses, Marco Cosentino Lagomarsino, Marco Lorenzi, Yacine Ahmine, Ville Mustonen, Miguel Correa Marrero and Agnese Seminara for discussions and critical reading of the manuscript. This work was supported by Agence Nationale de la Recherche (ANR-11-LABX-0028-01, ANR-15-IDEX-01, ANR-18-CE12-0004, ANR-20-CE12-0020, ANR-22-CE12-0015, ANR-24-CE12-7740), Fondation pour la Recherche Médicale (EQU202003010413), Impulscience - Fondation Bettencourt Schueller, UCA AAP Start-up Deep tech, CEFIPRA, SATT Sud-Est to GL. This project received funding from the European Union's Horizon 2020 research and innovation programme under grant agreement No 847581, the Region Sud, and the UCA JEDI.

## Author contributions

**Sakshi Khaiwal**: Conceptualization; Data curation; Software; Formal analysis; Validation; Investigation; Visualization; Methodology; Writing—original draft; Writing—review and editing. **Matteo De Chiara**: Conceptualization; Data curation; Formal analysis; Supervision; Investigation; Methodology. **Benjamin P Barré**: Investigation; Methodology. **Inigo Barrio-Hernandez**: Conceptualization; Formal analysis; Investigation; Methodology. **Simon Stenberg**: Formal analysis; Investigation; Methodology. **Pedro Beltrao**: Conceptualization; Funding acquisition; Methodology. **Jonas Warringer**: Conceptualization; Data curation; Funding acquisition; Methodology. **Gianni Liti**: Conceptualization; Resources; Supervision; Funding acquisition; Writing—original draft; Project administration; Writing—review and editing.

Source data underlying figure panels in this paper may have individual authorship assigned. Where available, figure panel/source data authorship is listed in the following database record: biostudies:S-SCDT-10_1038-S44320-025-00136-y.

## Disclosure and competing interests statement

The authors declare no competing interests.

# Expanded View Figures

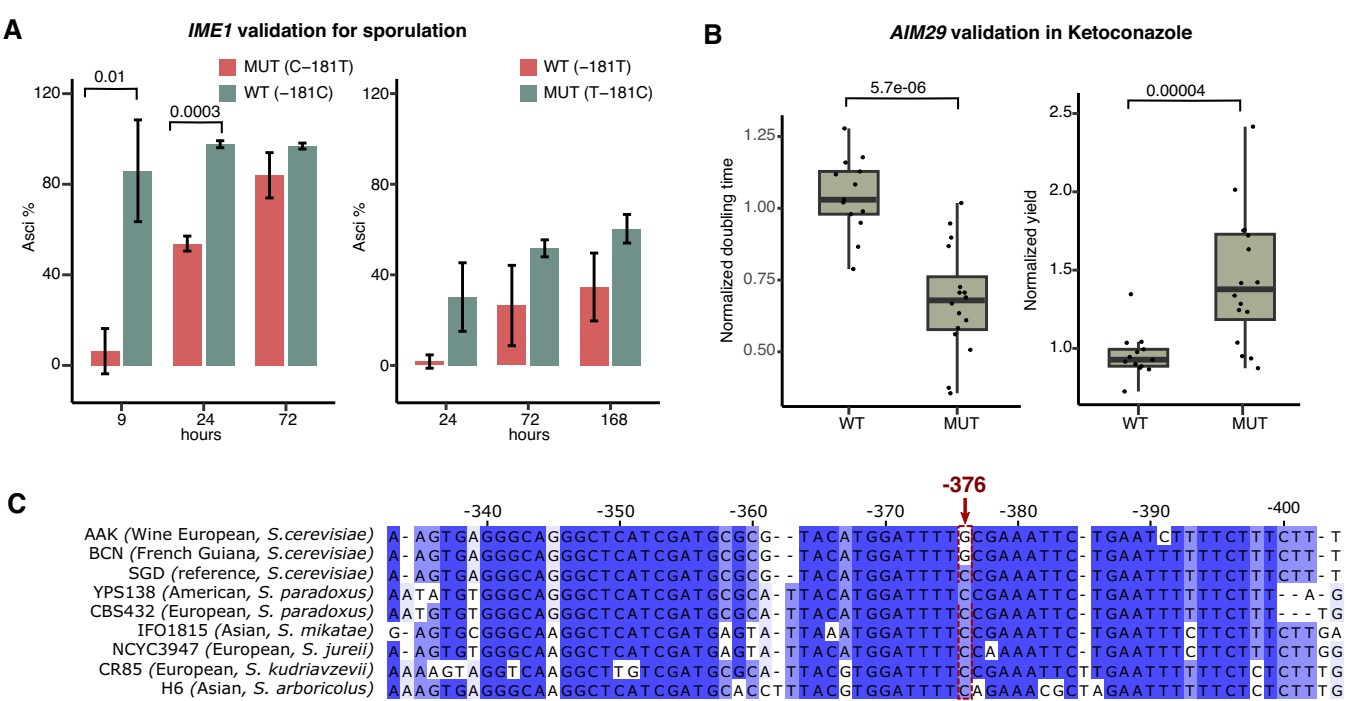

**Figure EV1.  Highly associated variants.**

Panels (**A**, **B**) show experimental validation of the GWAS SNPs in *IME1* promoter and *AIM29* gene respectively. (**A**), Percentage of asci production for two genetic-backgrounds: in the high sportulator strain YPS128 strain (left panel) the native promoter (WT -181C) was edited to the poor sporulating variant (MUT C-181T), resulting in significant difference in asci production at 9 and 24 h (t test, error bars indicate standard deviation across 3 replicates). In the poor sporulating strain YJS5833_CIA the native promoter (WT -181T) was edited to the high sporulating variant (MUT T-181C), resulting only in a mild increase in sporulation efficiency, underlying how the single variant effect is shaped by interactions with the genomic background (data replotted from (De Chiara et al, 2022)). (**B**) Growth rate and growth yield in ketoconazole (0.0075 μg/ml) showed significant differences between the BY4743 WT *AIM29* allele (13 replicates) edited for the synonymous variant (MUT C159T, His53His) (16 replicates, Wilcoxon test), data from https://github.com/SakshiKhaiwal/CEFIPRA. The median (50th percentile) of the data is indicated by the black horizontal line in the boxplots, the lower and upper bound of the box indicates the 25th (Q1) and 75th (Q3) percentile, respectively. The whiskers (vertical lines) extends to the smallest and largest value within 1.5* interquartile range (Q3-Q1) and the points beyond the ends of the line are potential outliers. (**C**) Mutation in the promoter region of *SKN7* associated to ketoconazole resistance map within a highly conserved non-coding region of the *Saccharomyces* genus.

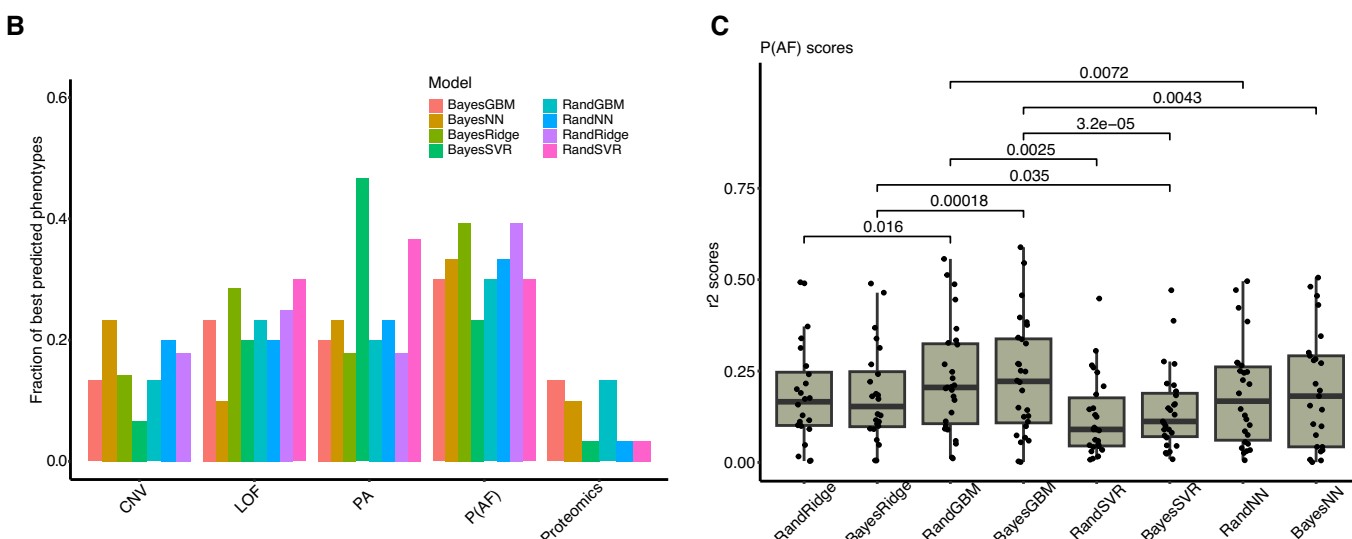

◀

**Figure EV2.   Benchmarking ML models on 30 test phenotypes.**

(A) Pairwise Pearson's correlation comparisons between the prediction accuracies from 8 ML strategies benchmarked using a subset of 30 phenotypes. The two hyperparameter optimization strategies, Bayesian (Bayes) and random (Rand), showed high correlation for each ML model. Moreover, Bayesian GBM showed higher similarity with both ridge and neural networks (NN) compared to random GBM, while SVR showed the highest deviation in the predictions compared to the rest of the models. (B) Fractions of phenotypes best predicted (calculated as the number of phenotypes that are best predicted by a given predictor compared to the rest out of all 30 tested phenotypes) with each predictor across 8 methods for the 30 phenotypes test set shows PA and P(AF) scores to be the top predictors. However, each predictor can predict a significant fraction of phenotypes. (C) Comparing the distributions of $r^2$ scores for the 30 phenotypes based on P(AF) scores showed GBM to be the best and SVR to be the worst prediction model. Significant differences between different methods that use the same hyperoptimization technique are reported (paired Wilcoxon test). The median (50th percentile) of the data is indicated by the black horizontal line in the boxplots, the lower and upper bound of the box indicates the 25th (Q1) and 75th (Q3) percentile respectively. The whiskers (vertical lines) extends to the smallest and largest value within 1.5* interquartile range (Q3-Q1) and the points beyond the ends of the line are potential outliers. Source data are available online for this figure.

 

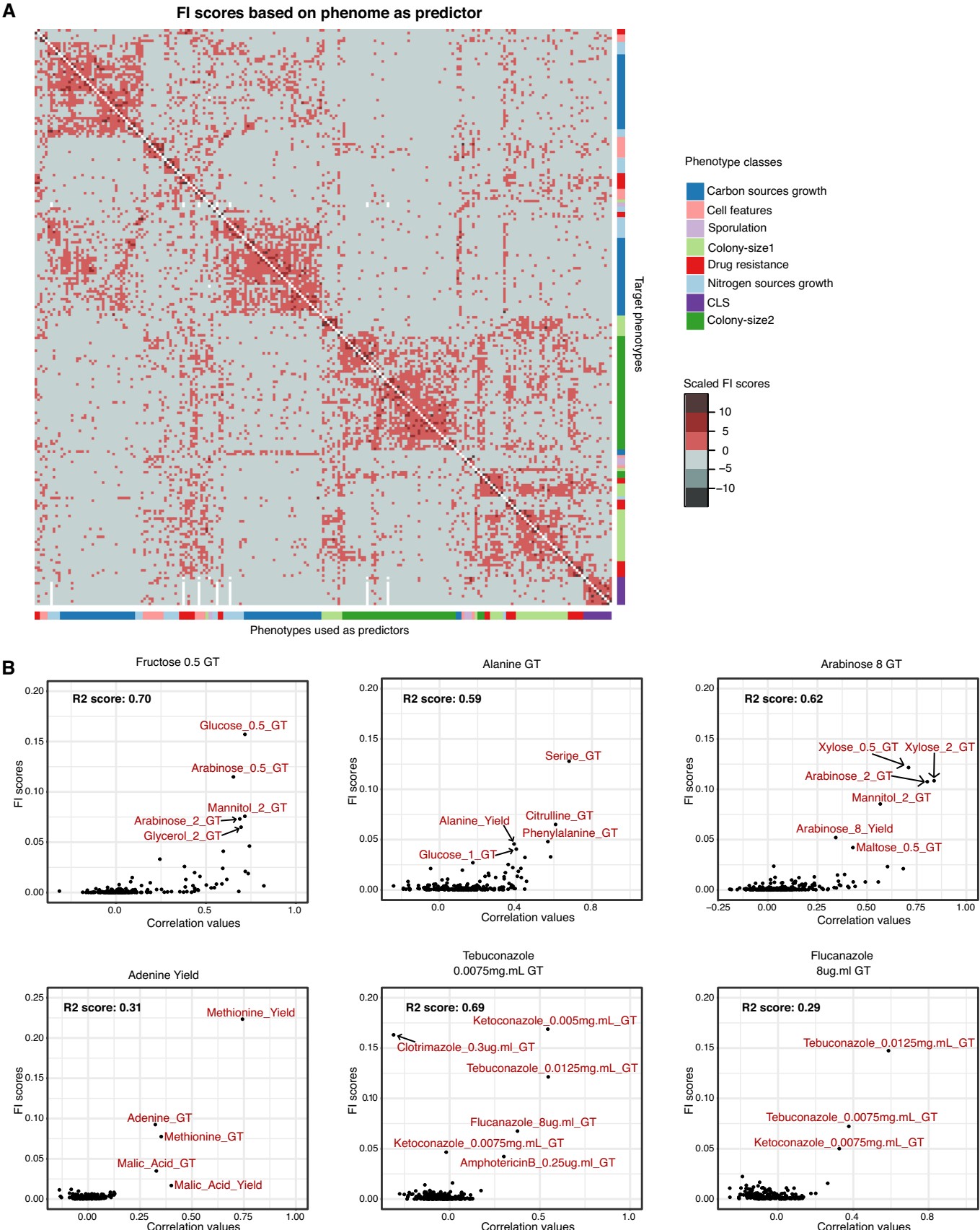

**A**　**FI scores based on phenome as predictor**

Phenotype classes

- Carbon sources growth
- Cell features
- Sporulation
- Colony-size1
- Drug resistance
- Nitrogen sources growth
- CLS
- Colony-size2

Scaled FI scores

**B**

**Figure EV3. Phenotype-based FI scores over phenotype correlations.**

(A) FI scores for all the phenotypes used as predictors for individual target phenotypes clustered according to phenotype correlations. This heat map shows that larger amounts of information are being shared between similar conditions and traits. (B) Examples showing that traits measured in similar conditions are more informative for prediction of growth rate or yield, than traits measured in identical conditions. The *x* axis shows the correlation values between the predicted phenotype (shown on top) and the rest, used as predictors, while the *y* axis depicts the feature importance scores. The $r^2$ scores from predictions are reported inside each box, while the top feature importance scores are highlighted in red.

                                                                           

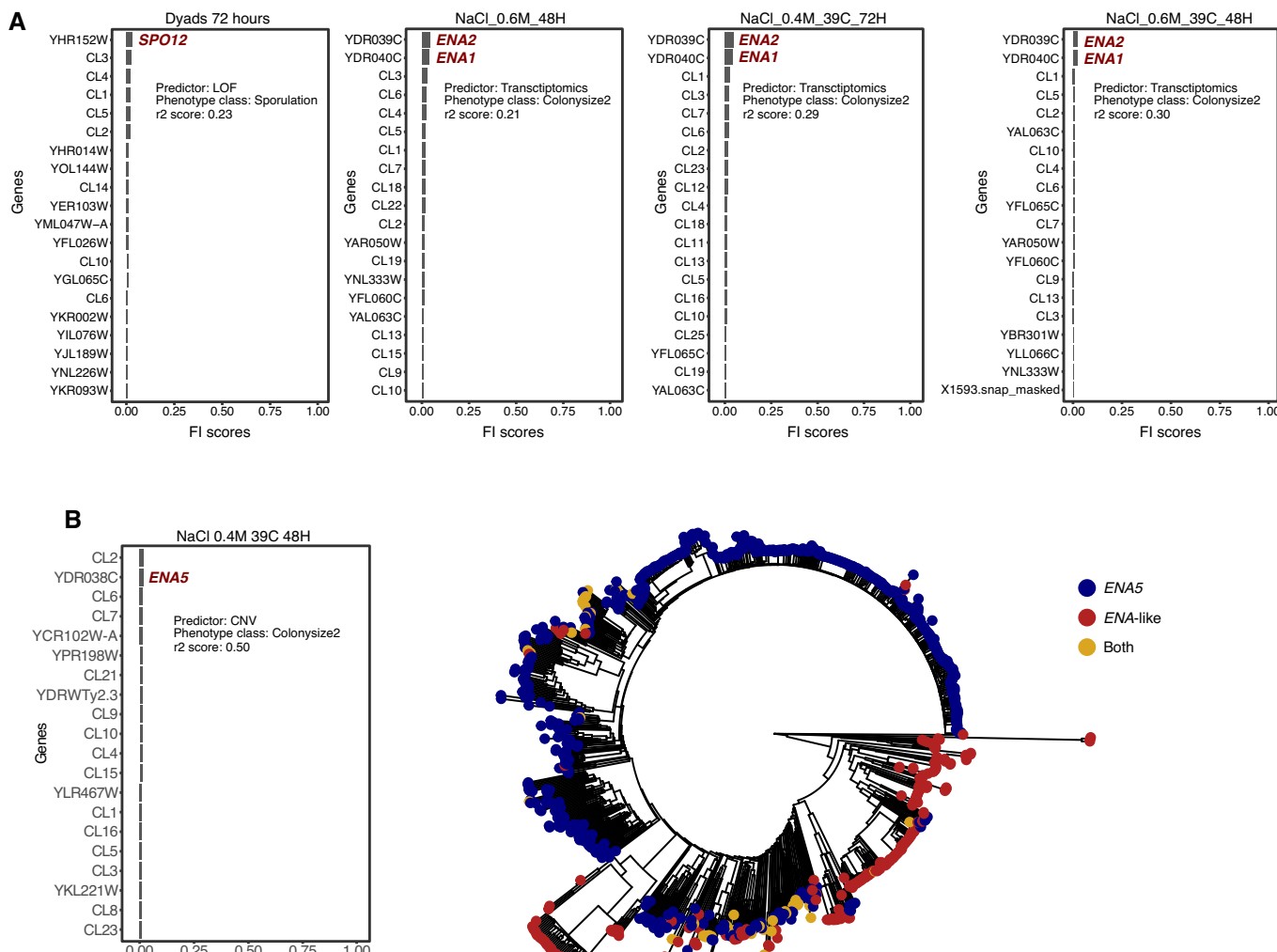

**Figure EV4.  ML identifies functionally relevant features impacting phenotypes.**

(**A**) *SPO12* and *ENA* genes are consistently the top predictor across different time points and concentrations in Dyads production and growth in NaCl respectively. (**B**) *ENA5* (De Chiara et al, 2022; Peter et al, 2018; D'Angiolo et al, 2023; Galardini et al, 2019) gene (representative of the reference-type ENA, which includes *ENA1, ENA2, ENA5* and *ENA6* in the *S. cerevisiae* pangenome (Peter et al, 2018)) is the top gene-predictor for growth measured in NaCl using copy number variation with population structure as the predictor. Population structure features also emerge as top FI scores (depicted by "CL"). This is consistent with the two major ENA alleles, reference-type and *ENA*-like annotated as VAR217 in the *S. cerevisiae* pangenome (Peter et al, 2018) (90% similarity to *ENA* reference), being clade-specific as illustrated by their distribution along the phylogenetic tree (phylogenetic tree, right panel). The phylogenetic tree was drawn and labeled using the Newick file from the 1,011 *S. cerevisiae* collection (Peter et al, 2018).

