## [Peer Review File · Molecular Systems Biology]

Predicting natural variation in the yeast phenotypic landscape with machine learning

Sakshi Khaiwal, Matteo Chiara, Benjamin Barre, Inigo Barrio-Hernandez, Simon Stenberg, Pedro Beltrao, Jonas Warringer, and Gianni Liti

Corresponding author(s): Gianni Liti (gianni.liti@unice.fr) , Sakshi Khaiwal (sakshi.KHAIWAL@univ-cotedazur.fr)

Review Timeline:

Submission Date:	24th Oct 24
Editorial Decision:	11th Dec 24
Revision Received:	30th Apr 25
Editorial Decision:	26th May 25
Revision Received:	25th Jun 25
Accepted:	16th Jul 25

Editor: Jingyi Hou

Transaction Report:

11th Dec 2024

Manuscript Number: MSB-2024-12723

Title: Predicting the natural yeast phenotypic landscape with machine learning

Author: Sakshi Khaiwal

Matteo Chiara

Benjamin Barre

Inigo Barrio-Hernandez

Simon Stenberg

Pedro Beltrao

Jonas Warringer

Gianni Liti

Dear Dr. Liti,

Thank you for submitting your work to Molecular Systems Biology. We have now heard back from the three reviewers who agreed to evaluate your manuscript. As you will see from the reports below, the reviewers acknowledge the interest of the study. They raise, however, a series of concerns, which we would ask you to address in a major revision.

While Reviewers #1 and #2 are generally supportive, Reviewer #3 raised several significant concerns. Specifically, Reviewer #3 noted that the current manuscript lacks precision and detail in the methods, pointed out several issues related to data analysis, and mentioned that the novel insight should be more clearly highlighted. These concerns should be carefully addressed.

As you may already know, our editorial policy allows in principle a single round of major revision, and it is therefore essential to provide responses to the reviewers' comments that are as complete as possible. Please feel free to contact me in case you would like to discuss in further detail any of the issues raised by the reviewers.

On a more editorial level, we would ask you to address the following issues:

- Please provide a .docx formatted version of the manuscript text (including legends for main figures, EV figures and tables). Please make sure that the changes are highlighted to be clearly visible.
- Please provide individual production quality figure files as .eps, .tif, .jpg (one file per figure).
- Please provide a .docx formatted letter INCLUDING the reviewers' reports and your detailed point-by-point responses to their comments. As part of the EMBO Press transparent editorial process, the point-by-point response is part of the Review Process File (RPF), which will be published alongside your paper.
- Please note that all corresponding authors are required to supply an ORCID ID for their name upon submission of a revised manuscript.
- We replaced Supplementary Information with Expanded View (EV) Figures and Tables that are collapsible/expandable online (see examples in <http://msb.embopress.org/content/11/6/812>). A maximum of 5 EV Figures can be typeset. EV Figures should be cited as 'Figure EV1, Figure EV2' etc... in the text and their respective legends should be included in the main text after the legends of regular figures.

Additional Tables/Datasets should be labeled and referred to as Table EV1, Dataset EV1, etc. Legends have to be provided in a separate tab in case of .xls files. Alternatively, the legend can be supplied as a separate text file (README) and zipped together with the Table/Dataset file.

For the figures and tables that you do NOT wish to display as Expanded View figures, they should be bundled together with their legends in a single PDF file called *Appendix*, which should start with a short Table of Content. Each legend should be below the corresponding Figure/Table in the Appendix. Appendix figures and tables should be referred to in the main text as: "Appendix Figure S1, Appendix Figure S2, Appendix Table S1" etc. See detailed instructions regarding expanded view here: <https://www.embopress.org/page/journal/17444292/authorguide#expandedview>.

- Before submitting your revision, primary datasets (and computer code, where appropriate) produced in this study need to be deposited in an appropriate public database (see <http://msb.embopress.org/authorguide-dataavailability> <https://www.embopress.org/page/journal/17444292/authorguide#dataavailability>). Please remember to provide a reviewer password if the datasets are not yet public. The accession numbers and database should be listed in a formal "Data Availability" section (placed after Materials & Method) that follows the model below (see also <https://www.embopress.org/page/journal/17444292/authorguide#dataavailability>). Please

note that the Data Availability Section is restricted to new primary data that are part of this study.

Data availability

-At EMBO Press we ask authors to provide source data for the main figures. Our source data coordinator will contact you to discuss which figure panels we would need source data for and will also provide you with helpful tips on how to upload and organize the files.

- Our journal encourages inclusion of *data citations in the reference list* to directly cite datasets that were re-used and obtained from public databases. Data citations in the article text are distinct from normal bibliographical citations and should directly link to the database records from which the data can be accessed. In the main text, data citations are formatted as follows: "Data ref: Smith et al, 2001". In the Reference list, data citations must be labeled with "[DATASET]". A data reference must provide the database name, accession number/identifiers and a resolvable link to the landing page from which the data can be accessed at the end of the reference. Further instructions are available at .

- We updated our journal's competing interests policy in January 2022 and request authors to consider both actual and perceived competing interests. Please review the policy <https://www.embopress.org/competing-interests> and update your competing interests if necessary.

Please use the heading "Disclosure statement and competing interests".

- All Materials and Methods need to be described in the main text using our 'Structured Methods' format. According to this format, the Methods section includes a Reagents and Tools Table (listing key reagents, experimental models, software and relevant equipment and including their sources and relevant identifiers) followed by a Methods and Protocols section describing the methods, ideally using a step-by-step protocol format. The aim is to facilitate adoption of the methodologies across labs.

Please download and fill our Reagents and Tools Table template (.docx), which you can find in our author guidelines:

<https://www.embopress.org/page/journal/17444292/authorguide#structuredmethods>.

- Regarding data quantification:

Please ensure to specify the name of the statistical test used to generate error bars and P values, the number (n) of independent experiments (please specify technical or biological replicates) underlying each data point and the test used to calculate p-values in each figure legend. Discussion of statistical methodology can be reported in the materials and methods section, but figure legends should contain a basic description of n, P and the test applied.

Graphs must include a description of the bars and the error bars (s.d., s.e.m.).

- Please provide a "standfirst text" summarizing the study in one or two sentences (approximately 250 characters, including space), three to four "bullet points" highlighting the main findings and a "synopsis image" (550px width and 400-600 px height, PNG format) to highlight the paper on our homepage.

Here are a couple of examples:

<https://www.embopress.org/doi/10.15252/msb.20199356>

<https://www.embopress.org/doi/10.15252/msb.20209475>

<https://www.embopress.org/doi/10.15252/msb.209495>

When you resubmit your manuscript, please download our CHECKLIST (<https://www.embopress.org/pb-assets/embo-site/EMBO%20Press%20Author%20Checklist-1642513524327.xlsx>) and include the completed form in your submission.

Please note that the Author Checklist will be published alongside the paper as part of the transparent process (<https://www.embopress.org/page/journal/17444292/authorguide#transparentprocess>).

If you feel you can satisfactorily deal with these points and those listed by the referees, you may wish to submit a revised version of your manuscript. Please attach a covering letter giving details of the way in which you have handled each of the points raised by the referees. A revised manuscript will be once again subject to review and you probably understand that we can give you no guarantee at this stage that the eventual outcome will be favorable.

I look forward to receiving your revised manuscript soon.

Kind regards,
Jingyi

Jingyi Hou, PhD
Senior Editor
Molecular Systems Biology

We realize that it is difficult to revise to a specific deadline. In the interest of protecting the conceptual advance provided by the work, we recommend a revision within 3 months (11th Mar 2025). Please discuss the revision progress ahead of this time with the editor if you require more time to complete the revisions.

IMPORTANT: When you send your revision, we will require the following items:

1. the manuscript text in LaTeX, RTF or MS Word format
2. a letter with a detailed description of the changes made in response to the referees. Please specify clearly the exact places in the text (pages and paragraphs) where each change has been made in response to each specific comment given
3. three to four 'bullet points' highlighting the main findings of your study
4. a short 'blurb' text summarizing in two sentences the study (max. 250 characters)
5. a 'thumbnail image' (550px width and max 400px height, Illustrator, PowerPoint or jpeg format), which can be used as 'visual title' for the synopsis section of your paper.
6. Please include an author contributions statement after the Acknowledgements section (see <https://www.embopress.org/page/journal/17444292/authorguide>)
7. Please complete the CHECKLIST available at (<https://bit.ly/EMBOPressAuthorChecklist>). Please note that the Author Checklist will be published alongside the paper as part of the transparent process (<https://www.embopress.org/page/journal/17444292/authorguide#transparentprocess>).
8. When assembling figures, please refer to our figure preparation guideline in order to ensure proper formatting and readability in print as well as on screen:
<https://bit.ly/EMBOPressFigurePreparationGuideline>
See also figure legend guidelines: <https://www.embopress.org/page/journal/17444292/authorguide#figureformat>
9. Please note that corresponding authors are required to supply an ORCID ID for their name upon submission of a revised manuscript (EMBO Press signed a joint statement to encourage ORCID adoption). (<https://www.embopress.org/page/journal/17444292/authorguide#editorialprocess>)
Currently, our records indicate that the ORCID for your account is 0000-0002-2318-0775.

Please click the link below to modify this ORCID:
Link Not Available

11. Include a Reagents and Tools Table as part of the Methods section, which can be downloaded from our author guidelines (<https://www.embopress.org/page/journal/17444292/authorguide#structuredmethods>)

*** PLEASE NOTE *** As part of the EMBO Press transparent editorial process initiative (see our Editorial at <https://dx.doi.org/10.1038/msb.2010.72>), Molecular Systems Biology publishes online a Review Process File with each accepted manuscripts. This file will be published in conjunction with your paper and will include the anonymous referee reports, your point-by-point response and all pertinent correspondence relating to the manuscript. If you do NOT want this File to be published, please inform the editorial office at msb@embo.org within 14 days upon receipt of the present letter.

Reviewer #1:

The study by Khaiwal and colleagues develops a machine learning framework for evaluating the relationship between genetic and phenotypic variation in a one-of-a-kind dataset of 1,011 genomes and 223 traits from the baker's yeast *Saccharomyces cerevisiae*. Although densely packed with analyses and data, the manuscript is generally clearly written and very informative. I also found that the manuscript's results were appropriately discussed and generally not over-interpreted, which is sometimes hard to do in these large-scale analyses. A straightforward-to-fix weakness of the study is the need for some clarifying details on the methods and clearer writing in certain parts, which could really help the readability of this paper (these have been noted in the minor comments section of the article).

Major comments:

The section describing the comparison of different ML models (pages 10-11) felt as a side note to the rest of the manuscript - given that all models seem to behave relatively similarly, my suggestion would be to relegate this part to the supplement and streamline the main text to the discussion of the biology. Alternatively, I note that which explanation of ML pre-processing is very thorough, especially for being in the results and not the methods section, explanation of what the different models are and how they are theoretically different is lacking in the results (but very detailed in the methods), so you could perhaps expand on this. I leave that decision of how to handle this section to the authors (but wanted to let them know that I felt this generated a pause in the flow of the rest of the manuscript).

The authors equate the term "yeast" with "*S. cerevisiae*", which is understandable but incorrect. There are many yeast species, both in the lineage that includes *S. cerevisiae* as well as in other lineages (e.g., the basidiomycete *Cryptococcus* is a yeast). This leads to incorrect statements such as that in page 3, line 23-24: "However, an in-depth investigation of whether ML enables higher resolution of the yeast genotype-to-phenotype is currently lacking." There are numerous papers from the Y1000+ Project that have recently employed ML approaches in the *Saccharomycotina* yeast subphylum that should be cited here (e.g., <https://doi.org/10.1073/pnas.2315314121>, <https://doi.org/10.1126/science.adj4503>, and <https://doi.org/10.1371/journal.pbio.3002832>). In addition, it would be useful if you discussed the distinction between examining the relationship between genotypic and phenotypic variation at the microevolutionary level (this study) vs. at the macroevolutionary level (studies from the Y1000+ project).

Minor comments:

Title page: the word "natural", without further description of what that means, in the title is a bit confusing. Could you please clarify in the abstract at least (if not the title) that these strains are isolated from the natural environment (Page 3, line 16: "ecological sources") as this is a strength of the paper?

Page 3, line 23-24: "However, an in-depth investigation of whether ML enables higher resolution of the yeast genotype-to-phenotype is currently lacking." Could check out Harrison et al. 2024!

Page 2, line 8: "pangenome" - it wasn't clear to me what you meant by this until I read the manuscript and realized that you refer to variation in the presence/absence of accessory genes. Consider replacing "pangenome" with the longer explanation.

Page 4, lines 8-9: Having trouble figuring out exactly what this means. "Overall, trait type had a stronger impact than environment type on trait similarity, with e.g. cell yields in different environments grouped together (Fig. 1c)". Can you please rephrase?

Page 4, line 12: explain what r/K means - most readers of MSB will be unfamiliar with the term

Figure 1: Clearer acronyms (or definitions in the figure/figure legend) would substantially help readability (CLS, CHX, etc).

Figure 1b:

- why the rows "starvation/caloric restriction" and "Chemicals/Heat/Drugs" appear in duplicate?
- Why do the numbers add up to 221 traits and not 223?
- What do the light blue, dark blue, and red rectangles mean? This should be explained in the legend

Figure 1c:

- Are the data used to make this figure available as a supplementary file?

Page 4, lines 28-33: When talking about the gene deletion phenotype library, are you saying that when genes are deleted, two traits tend to co-occur similarly to their correlation in their analyses? Having trouble understanding this paragraph - the general reader of MSB will not be familiar with the yeast gene deletion library data, so it would be good to revise so that the meaning of these analyses is more accessible to the general reader.

Page 5, lines 5-6: "Overall, comparing phenomic and genetic distances, either based on SNPs or gene presence-absence, we found no correlation." Please specify the statistical test performed and its results to support this statement.

Could highlight parts of figure 2c that are interesting/why, as it is it's a little hard to interpret.

Page 12, section title, "Near perfect prediction accuracy using phenomics data"

Potentially an overstatement of the fact that correlated phenotypes can predict each other very well-potentially could talk more about which phenotypes are correlated and do some type of clustering analysis?

Figure 4b could be discussed in more depth-what tends to be better predicted by genomic information? This could be a really interesting takeaway.

Page 14 line 2: It should be stated in the results what type of feature importance they are extracting (Gini impurity? Permutation?). Having trouble even finding it in methods.

I like all the information in Figure 5, however I would add how accurate the predictions are in 5b, and it's an interesting approach to average the feature importance for all predictions regardless of accuracy.

Page 16, lines 28-29: "These results mimic recent studies on bacteria antibiotic resistance, where pangenome variation is of even greater magnitude" - they also mimic the results of macroevolutionary studies of different yeast species abilities' to grow on different substrates since most of the variation is explained by variation in gene presence /absence (e.g., <https://doi.org/10.1073/pnas.2315314121>).

Reviewer #2:

Summary

The work of Khaiwal et al. builds upon the previous generation of an impressive genotype-phenotype library of 1011 diverse yeast strains. The authors first expand their dataset to include more phenotypic screenings. Then they perform correlation analyses to identify patterns in their dataset and uncover potential sources of co-correlation, followed by GWAS to identify important genetic predictors for the observed phenotypes.

In the crux of their work, the authors employ their dataset to make a first-of-its-kind comparison between the (by now conventional) GWAS approach, and alternative machine learning approaches. They first show promising results by obtaining reasonable predictions of phenotype from genotype data. Then they show machine learning's potential as a new GWAS tool, by using it to identify important (genotypic) features in their dataset that are linked to the corresponding phenotype, which they confirm with literature or follow-up experiments. One of their key findings is machine learning's ability to better grasp the importance of rare variants, a classical weakness of conventional GWAS studies.

General Remarks

In general, the work is exhaustive and the authors clearly did their best to cover all their bases. The authors amassed an impressive dataset, making this the first work of this scale comparing GWAS with alternative models. As such, it is an important first step in a next generation of genomic models that overcome some of the key weaknesses of conventional GWAS. The results are of interest to the entire genetic community, as the methodology is easily applicable to other organisms such as bacteria or human research.

It is clear that the authors put effort into shaping their research into a nice and concise story. The manuscript is a delight to read and the main messages are clear from the first read-through.

Major points

I have no major critiques or comments on this work.

Minor points

1. I noticed that scaling is performed separately on the training and test datasets. A better strategy is to perform scaling on both sets with the measures of the training set only. While this still avoids "data leakage" from the test-set to the training set, it allows for more fair model evaluation (and in all likelihood will increase your R-scores).

The issue with your current approach is that the test set might be distributed differently. If there is a high-value outlier in the test set, scaling will thus shift all values of the test set down and compress them relative to the training set, and will simultaneously make the outlier appears less extreme than it should. Ideally, we want identical values in the training and test set to be scaled identically, so the model can make accurate predictions.

To correctly implement this, the code can be modified very easily (edits in bold):

```
scaler = StandardScaler().fit(X_train.values)
X_train = pd.DataFrame(scaler.transform(X_train.values), columns=X_train.columns,
index=X_train.index)
```

```
X_test = pd.DataFrame(scaler.transform(X_test.values), columns=X_test.columns,
index=X_test.index)
```

2. Did the authors test different imputation treatments (eg. regression imputation with the sklearn IterativeImputer, or KNN imputation)? These might lead to better imputation than using the mean, since they use surrounding genetic information - although this could also hide interesting outliers. Alternatively, did the authors consider models that naturally handle missing values, such as HistGradientBoostingRegressor or RandomForest?
3. Page 7, Lines 26-29. It would increase the comprehension of the text if the authors added the name of the test and the p-values, instead of simply writing "exceptionally high p-values". Moreover, the following part about the experimental validation needs to be reformatted for increased clarity. Specifically, it needs to be clearer that only IME1 was validated in De Chiara et al 2022. Moreover, it is not clear whether SKN7 was also validated or not.
4. Figure 2:
 - (Optional) It would be more clear that the bottom panel is a zoom in of the line bar above if lines were added
 - I find the use of three colors for the different chromosomes confusing, I now expect the color to have meaning. It would be more clear that they are dividing the chromosomes if you only used two colors, or two shades of a single color. Alternatively, spacing could be added to separate the chromosomes, without the need for colors.
 - Since 3/6 types in the pie chart are invisible, I don't think it is adding information. Instead, the difference in numbers is more apparent from the amount of dots in the boxplots below. Why is "splice region" not included among the boxplots?
 - It is not immediately clear that the GWAS results from Fig. 2c are different from Supplementary Fig. 4c because a different number of SNP's was used. I would clarify this in (at least) one of the figure legends.
5. Figure 3c: If it does not make the figure too busy, it might be good to add lines connecting the dots between modeling strategies. This would visualize whether the same phenotypes are consistently predicted well/poorly (horizontal lines) or if some models are better at predicting some phenotypes (crossed lines), like in supplementary Fig. 8.
6. One of the difficulties of the author's dataset is the large diversity of yeast phenotypes. I am convinced that their approach would likely lead to even better results for mammalian genetics, where there is less diversity. This is something the authors can emphasize a bit more in the discussion, since it makes their work more meaningful and more likely to be adopted.
7. Given that the authors pipeline starts from LASSO or GWAS for feature selection, which are both linear methods, it is possible that some important interaction terms are removed and missed in the analysis. This could in future work hamper the ability to detect complex genetic interactions. I suggest the authors briefly mention this in their discussion, to inform readers that want study interactions with this approach. Non-linear feature selection also exist, like Boruta or BorutaSHAP.
8. Page 20 line 27: I believe "mixture" should be "mixed"

Reviewer #3:

This report was co-written by three experts.

The manuscript entitled "Predicting the natural yeast phenotypic landscape with machine learning" describes a quantitative genetics study where machine-learning (ML) approaches are used to predict phenotypes of wild yeast strains. The authors first assembled a large dataset of phenotypes of the 1,011 yeasts collection. This dataset regroups previously-described data (190 traits) with 33 additional traits that they quantified in the present study. They describe the structure of phenotype-phenotype correlations. They conducted a genetic association study (GWAS) and they describe the properties of genetic linkages inferred. The authors then grouped strain-level information of phenotypes, genotypes and molecular traits (transcriptome, proteome) to train various types of ML models and evaluate their performance to predict phenotypes. After reducing the number of informative features using LASSO-based selection, they trained models on 75% of data and evaluated them using the remaining 25% data. They report that Gradient Boosted Machines performed best, that transcriptome and gene presence/absence were the worse and best molecular predictor, respectively, and that including SNPs data did not improve predictions. They also report that including some phenotypes as features drastically increase prediction power. Finally, they provide interpretations and discussions on specific genes detected as molecular predictors of one or more phenotypes.

Although the quantity of data analyzed is impressive, we consider that the study severely suffers from major issues.

1. There is a general lack of precision and details on the methods that were used and on the results that are presented. In particular:
 - Data quality and precision of measurements may vary among traits (supp data 4 shows experimental error, but not intra-strain biological variability), which could be an important factor contributing to trait predictability. This should be reported and discussed in the manuscript.

- Traits may not all show the same degree of variation among strains, which could also contribute to trait predictability. It seems important to show and discuss how trait variation among strains is related to trait predictability.
- "The data was scaled using standard scaling" (p21) is too imprecise to understand what was done.
- It is unclear what information is used about each predictor for training the model. For example in the case of CNVs, does the model use information about genomic location of CNVs, their length, the number of genes included, the number of copies? Similarly, what information is used for SNPs?

How are LOF variants defined and why do they not overlap with SNPs (as shown in Figure 2a)?

- Trait-trait correlations are quantified but their statistical significance is not analyzed. The sentence of line 5 of page 4 "we found significant correlations to be both..." is therefore wrong because the authors do not know which correlations are significant. Similarly, no statistical test is provided to support the correlation between azole and CLS traits.
- line 24 of p4 mentions "hierarchical clustering with k=4": did the authors apply k-means clustering? Or did they cut a hierarchical tree?
- There are several instances where a direct validation of genetic effects is mentioned, without showing the results. AIM29 (line 27, p7): The results are accessible on a github page and mentioned as "manuscript in prep", while simultaneously citing ref 17. What does this mean? Simply show the results in the present manuscript or please don't mention them. SKN7 (p7): where are the results? For SEC11: Supp Data 12 mentions "Unpublished data from <https://www.nature.com/articles/s41559-022-01671-9>". What does this mean? Published or unpublished? Supp Data 14 and 15 contain results (p-values) that cannot be interpreted because what was done from SuppData12 is not written. SPO12 is said to be validated (p14) from ref 17: what validation exactly? Where in ref17? Line 22 p 16 "SNPs... did not show detectable phenotypic effects when reconstructed in different backgrounds". Where is the data from these reconstructions? We simply don't know where to look for the results.

2. If the authors applied GWAS as they wrote in methods (line 29, p20), then they considered all linkage hits passing a 5% family-wise error rate as being significant and described the properties of these hits as an ensemble of results. This way, multiple testing is corrected for the number of genetic markers but not for the number of phenotypes, which is large here (n=223). This is a major problem because we do not know what would remain of Fig2 (and related text) if the authors had properly controlled for genome x phenome multiple testing. For example, the Manhattan plot of Fig2a shows many dots, even with very weak scores, as if they were all statistically significant (according to legend). If most hits are false positives, as we suspect, then discussing the absence of patterns (eg. pages 7,8) among them is useless. Also, a large fraction of false positives would be a simple explanation of why "GWAS hits also included equal proportions of synonymous and missense mutations" (page 16).

In addition, the authors seem to be using the term "SNPs" as ML predictors to refer to GWAS hits. If this is the case, then i) many of these hits are of poor quality (non-significant) and therefore discovering that they poorly predict traits is not a finding and ii) these features were selected (by GWAS) on the same dataset as the one used for model training, which is a "post-selection inference" issue (same as mentioned below for LASSO).

3. The novel knowledge produced by the study is somewhat difficult to discern. For example, the authors emphasize that machine learning showed that some traits are much better than other features (eg. molecular) to predict phenotypes (page 12). This is largely expected given the strong correlations present in the phenotypic matrix. What exactly was discovered in addition to simple expectation? Page 17 (line 8) writes that the study "suggests that predicting complex traits based on genomic information in natural populations is feasible". This is too vague, readers already know that quantitative genetics is feasible. In addition, other studies used ML to detect rare variants associated with phenotypic variation and should be cited (eg, Petrazzini et al., 2024 <https://doi.org/10.1038/s41588-024-01791-x>)

4. LASSO-based reduction of features preceding machine learning seems to be applied on the same dataset as downstream model trainings. This is an issue called "post-selection inference" that is known to introduce potential biases in the conclusions. Rather, the authors should a minima separate the data into training and test sets before LASSO selection. The subject should be mentioned and discussed in the paper. References can be found in Kuchibhotla et al, 2022 (<https://doi.org/10.1146/annurev-statistics-100421-044639>).

5. The way prediction accuracy is evaluated is not appropriate. Person's correlation coefficients between empirical and predicted values informs on the global ability of a model to predict a set of trait values, but not on how accurate each prediction is. The authors should conduct other performance evaluation and provide other standard metrics such as prediction error.

6. The fact that transcriptomics is a poor predictor would be a very intriguing (and therefore interesting) result if the authors showed that the quality of the transcriptomics dataset is high. Otherwise, poor predictive power may simply result from poor quality.

Finally, ML is presented here as if it were a standard, homogeneous approach (e.g. page 3, lines 23-27) but it is an entire scientific field. The authors should specify that they used "some Machine Learning approaches" (not all existing ones) "in a standard procedure". There surely exists other ML methods that will perform better or worse in this context. Also, the "Deep Neural Networks" paragraph on page 23 describes what seems to be a specific type of neural networks, i.e. dense (or fully connected), but a wider range of "deep neural networks" exist and this should be mentioned.

Specific comments:

- abstract: "multiple linear and complex machine learning methods". A linear ML method can be complex and non-linear methods can be simple.
- abstract: The sentence "We determined gradient boosting machines as best performing in 65% of predictions and pangenome as best predictor" is totally unclear, especially regarding the link between the cited method performance and the biological finding.
- abstract: "($r = 0.2-0.9$)" r is not defined (and this information is maybe too specific for an abstract).
- line12, p4, the letter 'r' is used to mention too different things in the same sentence.
- line13, p4: "with that such"?
- line26, p7: "exceptionally high p-values": No, the $-\log_{10}$ of them is high, so the p-values are low.
- line 18, p8: PPR is mentioned here, but the methods describe "network expansion" instead. Please be consistent.
- what does the star on Fig1d represent? How does it "highlight" the connection between phenotypes measured in different studies? (line 17 p 4)?
- Figure 2b: Numbers above box plots probably correspond to p-values, but this should be made more explicit. How was SNPs weight (y-axis) calculated?
- line 17, p10: SVR is not defined. Is it the same thing as SVM defined in line 12?
- line 28, p10: NN is not defined.
- Figure 2c: The plot is difficult to read because phenotypes are only shown on y-axis and not shown on x-axis (Figure 1c is easier to read).
- Figure 3c: Did the authors use data points (black dots) shown on Figure 3c to perform Wilcoxon tests? BayesGBM and BayesSVR do not seem to possibly differ at P-value lower than 0.05.
- Figure 3d: Y-axis label "Fraction of best predicted phenotypes" differs from the legend "Fraction of the best performance...". What exactly is shown? Why is a pattern seen here and not in panel 3c?
- Figure 4a: Axes could be labeled more clearly. Why only 4 comparisons are shown among all possible combinations of 2 predictors?
- in methods, references should be provided for "PLINK v1.90b6.18", "the online web tool called Variant Effect Predictor (VEP)", "LASSO regression", "scikit-learn", "Hi-LASSO Python library 1.0.6", and all methods: ridge regression, elastic regression, support vector regression.

Response letter

We are thankful for the constructive feedback provided by the reviewers. We performed all the suggested analyses and revised our manuscript to address all reviewers' comments. We believe that these modifications have further strengthened our work. Details of our responses and the corresponding changes in the manuscript are provided below. The reviewers' comments are in black font, and our replies & actions are in blue font.

[R] - Response to the reviewer

[A] - Actions taken in the revised manuscript

Reviewer #1:

The study by Khaiwal and colleagues develops a machine learning framework for evaluating the relationship between genetic and phenotypic variation in a one-of-a-kind dataset of 1,011 genomes and 223 traits from the baker's yeast *Saccharomyces cerevisiae*. Although densely packed with analyses and data, the manuscript is generally clearly written and very informative. I also found that the manuscript's results were appropriately discussed and generally not over-interpreted, which is sometimes hard to do in these large-scale analyses. A straightforward-to-fix weakness of the study is the need for some clarifying details on the methods and clearer writing in certain parts, which could really help the readability of this paper (these have been noted in the minor comments section of the article).

Major comments:

1. The section describing the comparison of different ML models (pages 10-11) felt as a side note to the rest of the manuscript - given that all models seem to behave relatively similarly, my suggestion would be to relegate this part to the supplement and streamline the main text to the discussion of the biology. Alternatively, I note that which explanation of ML pre-processing is very thorough, especially for being in the results and not the methods section, explanation of what the different models are and how they are theoretically different is lacking in the results (but very detailed in the methods)., so you could perhaps expand on this. I leave that decision of how to handle this section to the authors (but wanted to let them know that I felt this generated a pause in the flow of the rest of the manuscript).

[R&A] We think the comparison between the different ML models and extensive benchmarking is quite significant to our study and deserves to be reported in the result section. However, we rebalanced the description of the preprocessing and ML models between results and method sections. Specifically, we shortened the description of the preprocessing in the results and further elaborated on this in the method section (page 20, lines 19-22 and 28-30) and added a short description of the four ML algorithms to the results section (page 6, lines 29-33).

2. The authors equate the term "yeast" with "*S. cerevisiae*", which is understandable but incorrect. There are many yeast species, both in the lineage that includes *S. cerevisiae* as well as in other lineages (e.g., the basidiomycete *Cryptococcus* is a yeast). This leads to incorrect statements such as that in page 3, line 23-24: "However, an in-depth investigation of whether ML enables higher resolution of the yeast genotype-to-phenotype is currently lacking." There are numerous papers from the Y1000+ Project that have recently employed ML approaches in the *Saccharomycotina* yeast subphylum that should be cited here (e.g., <https://doi.org/10.1073/pnas.2315314121>, <https://doi.org/10.1126/science.adj4503>, and

<https://doi.org/10.1371/journal.pbio.3002832>). In addition, it would be useful if you discussed the distinction between examining the relationship between genotypic and phenotypic variation at the microevolutionary level (this study) vs. at the macroevolutionary level (studies from the Y1000+ project).

[R&A] We agree that referring to *S. cerevisiae* as yeast in general is incorrect and have now edited the text in several places. We apologise for the sentence related to the lack of ML approaches applied to yeast genotype to phenotype map, which we intended to refer to the application of ML to quantitative phenotypic variation within species. We have now clarified this aspect both in the introduction (page 2, lines 26-28) and discussion (page 11, lines 10-17) and included the relevant references.

Minor comments:

Title page: the word "natural", without further description of what that means, in the title is a bit confusing. Could you please clarify in the abstract at least (if not the title) that these strains are isolated from the natural environment (Page 3, line 16: "ecological sources") as this is a strength of the paper?

[R&A] We agree that the word "natural" in the title was unclear and have now revised the title. We also clarified in the introduction that our collection is made of *S. cerevisiae* strains isolated world-wide (page 2, lines 17-19).

Page 3, line 23-24: "However, an in-depth investigation of whether ML enables higher resolution of the yeast genotype-to-phenotype is currently lacking." Could check out Harrison et al. 2024!

[R&A] We edited the text to better position our study in relation to the current state-of-the-art (page 2, lines 26-28). See also our reply to the major point 2 above.

Page 2, line 8: "pangenome" - it wasn't clear to me what you meant by this until I read the manuscript and realized that you refer to variation in the presence/absence of accessory genes. Consider replacing "pangenome" with the longer explanation.

[R&A] We edited the abstract to use presence/absence variation (page 1, lines 22-23) and defined what we refer with "pangenome" variation in the result section (page 4, lines 30-32).

Page 4, lines 8-9: Having trouble figuring out exactly what this means. "Overall, trait type had a stronger impact than environment type on trait similarity, with e.g. cell yields in different environments grouped together (Fig. 1c)". Can you please rephrase?

[R&A] We meant that trait-type (e.g. growth rates in different environments) tends to have higher correlations compared to different trait-type (e.g. growth rate and growth yield) measured in a given environment. We rephrased this sentence (page 3, lines 15-18) and edited Figure 1A-C labelling to further clarify this point.

Page 4, line 12: explain what r/K means - most readers of MSB will be unfamiliar with the term

[R&A] We agree that this sentence was unclear and have simplified it by referring to this as a trade-off and maintained reference 28, which discusses in depth the r/K concept (page 3, line 20-23).

Figure 1: Clearer acronyms (or definitions in the figure/figure legend) would substantially help readability (CLS, CHX, etc).

[R&A] We have now defined these acronyms in Figure 1 legend (page 14).

Figure 1b:

- why the rows "starvation/caloric restriction" and "Chemicals/Heat/Drugs" appear in duplicate?

[R&A] They were separated since they came from different studies, but we agree this might be confusing and have now joined them together in Figure 1B.

- Why do the numbers add up to 221 traits and not 223?

[R&A] Growth rate and yield in rich medium, i.e. YPD, were initially excluded from the phenotype classes. We have now included them in the carbon sources growth phenotype classes, as at least the growth yield is limited by carbon availability. Now all phenotypes reported in Figure 1B add to 223.

- What do the light blue, dark blue, and red rectangles mean? This should be explained in the legend

[R&A] This information has now been added to Figure 1B legend (page 14).

Figure 1c:

- Are the data used to make this figure available as a supplementary file?

[R&A] Yes, all the phenotypes and corresponding information are provided in Dataset EV1.

Page 4, lines 28-33: When talking about the gene deletion phenotype library, are you saying that when genes are deleted, two traits tend to co-occur similarly to their correlation in their analyses? Having trouble understanding this paragraph - the general reader of MSB will not be familiar with the yeast gene deletion library data, so it would be good to revise so that the meaning of these analyses is more accessible to the general reader.

[R&A] Yes, the reviewer's interpretation is correct. We have now clarified this section and added additional details and references to further introduce the *S. cerevisiae* deletion collection (page 4, lines 4-6 and page 18, lines 6-8 in methods).

Page 5, lines 5-6: "Overall, comparing phenomic and genetic distances, either based on SNPs or gene presence-absence, we found no correlation." Please specify the statistical test performed and its results to support this statement.

[R&A] We now added the corresponding Pearson's correlation coefficient along with the p-values for both distances, i.e. between phenotypes and SNPs and between phenotypes and gene presence-absence. In addition, we have also edited the sentence to specify that very weak correlations are observed between them (page 4, lines 16-18).

Could highlight parts of figure 2c that are interesting/why, as it is it's a little hard to interpret.

[R&A] We have now highlighted the results explained in the text (page 5, lines 18-23) in Figure 2C.

Page 12, section title, "Near perfect prediction accuracy using phenomics data"

Potentially an overstatement of the fact that correlated phenotypes can predict each other very well-potentially could talk more about which phenotypes are correlated and do some type of clustering analysis?

[R&A] We edited the title paragraph to better reflect the revised content of the paragraph and avoid any potential overstatement (page 7, line 24). We further investigated the relationship between phenotypic correlations and FI scores of phenotypes and indeed, the best predictions are coming from phenotypes in similar environments at different concentrations or time points. However, improved predictions are also seen in conditions with similar nutrient sources (for example carbon or nitrogen sources, drugs). These results are now included in the new Figure EV3 A-B and discussed in the results (page 8, lines 25-30).

Figure 4b could be discussed in more depth-what tends to be better predicted by genomic information? This could be a really interesting takeaway.

[R&A] We have now further discussed the results reported in Figure 4A (previously Figure 4b) (page 7, lines 27-34 and page 8, lines 1-7). Furthermore, we added a new panel (Figure 4B) to show a comparison between all pairwise combinations of predictors to show that most phenotypes (with a few outliers) are consistently either predicted well or poorly across predictors. We further expanded the related discussion (page 11, lines 26-33).

Page 14 line 2: It should be stated in the results what type of feature importance they are extracting (Gini impurity? Permutation?). Having trouble even finding it in methods.

[R&A] This is now briefly described in the results (page 7, lines 1-3) and further elaborated on in the section on "Feature importance calculation", which has now been added to the methods section of the GenPhen pipeline (page 23, lines 1-5).

I like all the information in Figure 5, however, I would add how accurate the predictions are in 5b, and it's an interesting approach to average the feature importance for all predictions regardless of accuracy.

[R&A] Thank you, we have now added the r^2 scores for the test set in Figure 5B.

Page 16, lines 28-29: "These results mimic recent studies on bacteria antibiotic resistance, where pangenome variation is of even greater magnitude" - they also mimic the results of macroevolutionary

studies of different yeast species abilities' to grow on different substrates since most of the variation is explained by variation in gene presence /absence (e.g., <https://doi.org/10.1073/pnas.2315314121>).

[R&A] We have now revised the text to discuss the recent studies on the Y1000+ dataset at the macroevolutionary level (pages 11, lines 10-17).

Reviewer #2:

Summary

The work of Khaiwal et al. builds upon the previous generation of an impressive genotype-phenotype library of 1011 diverse yeast strains. The authors first expand their dataset to include more phenotypic screenings. Then they perform correlation analyses to identify patterns in their dataset and uncover potential sources of co-correlation, followed by GWAS to identify important genetic predictors for the observed phenotypes.

In the crux of their work, the authors employ their dataset to make a first-of-its-kind comparison between the (by now conventional) GWAS approach, and alternative machine learning approaches. They first show promising results by obtaining reasonable predictions of phenotype from genotype data. Then they show machine learning's potential as a new GWAS tool, by using it to identify important (genotypic) features in their dataset that are linked to the corresponding phenotype, which they confirm with literature or follow-up experiments. One of their key findings is machine learning's ability to better grasp the importance of rare variants, a classical weakness of conventional GWAS studies.

General Remarks

In general, the work is exhaustive and the authors clearly did their best to cover all their bases. The authors amassed an impressive dataset, making this the first work of this scale comparing GWAS with alternative models. As such, it is an important first step in a next generation of genomic models that overcome some of the key weaknesses of conventional GWAS. The results are of interest to the entire genetic community, as the methodology is easily applicable to other organisms such as bacteria or human research.

It is clear that the authors put effort into shaping their research into a nice and concise story. The manuscript is a delight to read and the main messages are clear from the first read-through.

Major points

I have no major critiques or comments on this work.

Minor points

1. I noticed that scaling is performed separately on the training and test datasets. A better strategy is to perform scaling on both sets with the measures of the training set only. While this still avoids "data leakage" from the test-set to the training set, it allows for more fair model evaluation (and in all likelihood will increase your R-scores).

The issue with your current approach is that the test set might be distributed differently. If there is a high-value outlier in the test set, scaling will thus shift all values of the test set down and compress them relative to the training set, and will simultaneously make the outlier appears less extreme than it should.

Ideally, we want identical values in the training and test set to be scaled identically, so the model can make accurate predictions.

To correctly implement this, the code can be modified very easily (edits in bold):

```
scaler = StandardScaler().fit(X_train.values)
X_train = pd.DataFrame(scaler.transform(X_train.values), columns=X_train.columns,
index=X_train.index)
X_test = pd.DataFrame(scaler.transform(X_test.values), columns=X_test.columns,
index=X_test.index)
```

[R&A] Thank you for the suggestion. We have now implemented this in our pipeline, and all the prediction results are updated accordingly in Figures 3-5 and Supplementary Figures 7-10. This has led to a significant improvement in r^2 scores as illustrated by examples shown below using P(AF) scores as predictors and Bayesian optimized GBM model for the 223 phenotypes.

2. Did the authors test different imputation treatments (eg. regression imputation with the sklearn IterativeImputer, or KNN imputation)? These might lead to better imputation than using the mean, since they use surrounding genetic information - although this could also hide interesting outliers. Alternatively, did the authors consider models that naturally handle missing values, such as HistGradientBoostingRegressor or RandomForest?

[R&A] Based on the reviewer's suggestion, we tried KNN imputation, however, no difference in the prediction results was observed (shown below). This is also now implemented in the preprocessing part of the Gen-Phen and can be chosen by the user.

We had previously tested random forest, however these were computationally inefficient compared to GBM and did not improve the predictions. Moreover, we tried HistGradientBoostingRegressor for some test phenotypes, and again we did not see any improvements in predictions compared to GBM. Also,

HistGradientBoostingRegressor function in scikit-learn does not provide a direct way to extract feature importance scores and require implementation of additional permutation-based methods to extract them, hence for our purpose we stuck with GBM in the manuscript. However, HistGradientBoostingRegressor with Bayesian hyperparameters optimization is now provided as an additional model to be chosen for training in the Gen-Phen in the model section.

3. Page 7, Lines 26-29. It would increase the comprehension of the text if the authors added the name of the test and the p-values, instead of simply writing "exceptionally high p-values". Moreover, the following part about the experimental validation needs to be reformatted for increased clarity. Specifically, it needs to be clearer that only IME1 was validated in De Chiara et al 2022. Moreover, it is not clear whether SKN7 was also validated or not.

[R&A] We rephrased this paragraph (page 5, lines 6-9), added further details on these highly significant SNPs (Figure 2A legend, page 14 lines 21-22) and we added Figure EV1 to provide full reference of the sources of the validation experiments (see also reply to reviewer 3).

4. Figure 2:

- (Optional) It would be more clear that the bottom panel is a zoom in of the line bar above if lines were added

[R&A] This is now implemented in Figure 2A.

- I find the use of three colors for the different chromosomes confusing, I now expect the color to have meaning. It would be more clear that they are dividing the chromosomes if you only used two colors, or two shades of a single color. Alternatively, spacing could be added to separate the chromosomes, without the need for colors.

[R&A] We have changed Figure 2A and used two shades of a single color as suggested.

- Since 3/6 types in the pie chart are invisible, I don't think it is adding information. Instead, the difference in numbers is more apparent from the amount of dots in the boxplots below. Why is "splice region" not included among the boxplots?

[R&A] We improved this figure to increase the size of the pie chart and added black outlines to differentiate the different categories. The splice region SNPs do not appear as hits, which is most likely due to their low number in general (~200 in total out of ~460K). Hence, this category has now been removed.

- It is not immediately clear that the GWAS results from Fig. 2c are different from Supplementary Fig. 4c because a different number of SNP's was used. I would clarify this in (at least) one of the figure legends.

[R&A] We apologise if this aspect was unclear. In Figure 2C, we compared the correlation between phenotype pairs to shared variants significantly associated with these two phenotypes. In Supplementary Figure 4C, we showed the same patterns reflected at the gene level, where we mapped the variants to genes and compared the significantly associated genes between pairs of phenotypes. The legend of Supplementary Figure 4C has now been rephrased to clarify this aspect.

5. Figure 3c: If it does not make the figure too busy, it might be good to add lines connecting the dots between modeling strategies. This would visualize whether the same phenotypes are consistently predicted well/poorly (horizontal lines) or if some models are better at predicting some phenotypes (crossed lines), like in supplementary Fig. 8.

[R&A] Adding the lines to the former Figure 3C made it hard to read. To appreciate whether the same phenotypes are consistently predicted well/poorly, we have now replaced this with scatterplots to illustrate pairwise accuracy comparisons between the methods (Figure 3C and Figure EV2A) and moved the box plots to Figure EV2C in addition.

6. One of the difficulties of the author's dataset is the large diversity of yeast phenotypes. I am convinced that their approach would likely lead to even better results for mammalian genetics, where there is less diversity. This is something the authors can emphasize a bit more in the discussion, since it makes their work more meaningful and more likely to be adopted.

[R&A] We do agree with the reviewer's suggestion. This is also suggested by improved predictions in strains that are more closely related in the training and testing set than distantly related ones (Supplementary figure 15). Hence, we would expect our ML framework to perform better in higher eukaryotes such as humans, where the genetic diversity is lower than in yeast. We have now added this to the discussion (page 11, line 23-26).

7. Given that the authors pipeline starts from LASSO or GWAS for feature selection, which are both linear methods, it is possible that some important interaction terms are removed and missed in the analysis. This could in future work hamper the ability to detect complex genetic interactions. I suggest the authors briefly mention this in their discussion, to inform readers that want study interactions with this approach. Non-linear feature selection also exist, like Boruta or BorutaSHAP.

[R] We acknowledge that both these types of feature selection techniques are based on linear regression. However, the GBM method implemented in our pipeline is a non-linear method that reduces the number of features mimicking a feature selection criterion. We tried Boruta, which is based on a random forest algorithm, but found that it was computationally too inefficient to be used here. Therefore, using GBM is a more computationally efficient way of performing feature reduction based on decision trees. Hence, using a combination of no feature selection and GBM as the learning algorithm as parameters in our pipeline, we effectively build non-linear models with reduced features with no prior feature reduction using linear methods.

[A] We have now clarified this aspect in the results (Page 7, lines 25-27) and used this approach in all predictions for all phenotypes using different predictors (Figure 4-5). LASSO selection was used only when comparing all methods (Figure 3C-D and Figure EV2) and for predictions on SNPs to reduce training times.

8. Page 20 line 27: I believe "mixture" should be "mixed"

[R&A] This is now corrected.

Reviewer #3:

This report was co-written by three experts.

The manuscript entitled "Predicting the natural yeast phenotypic landscape with machine learning" describes a quantitative genetics study where machine-learning (ML) approaches are used to predict phenotypes of wild yeast strains. The authors first assembled a large dataset of phenotypes of the 1,011 yeasts collection. This dataset regroups previously-described data (190 traits) with 33 additional traits that they quantified in the present study. They describe the structure of phenotype-phenotype correlations. They conducted a genetic association study (GWAS) and they describe the properties of genetic linkages inferred. The authors then grouped strain-level information of phenotypes, genotypes and molecular traits (transcriptome, proteome) to train various types of ML models and evaluate their performance to predict phenotypes. After reducing the number of informative features using LASSO-based selection, they trained models on 75% of data and evaluated them using the remaining 25% data. They report that Gradient Boosted Machines performed best, that transcriptome and gene presence/absence were the worse and best molecular predictor, respectively, and that including SNPs data did not improve predictions. They also report that including some phenotypes as features drastically increase prediction power. Finally, they provide interpretations and discussions on specific genes detected as molecular predictors of one or more phenotypes.

Although the quantity of data analyzed is impressive, we consider that the study severely suffers from major issues.

1. There is a general lack of precision and details on the methods that were used and on the results that are presented. In particular:

- Data quality and precision of measurements may vary among traits (supp data 4 shows experimental error, but not intra-strain biological variability), which could be an important factor contributing to trait predictability. This should be reported and discussed in the manuscript.

[R&A] We thoroughly revised the methods section and added further details as detailed below and in our replies to the other reviewers comments. We have replicated measurements for 189 phenotypes out of 223. The replicates represent four experimental measurements on four different replicates of the same strain. Hence, the reported ‘standard error’ considers the intra-strain biological variability, i.e. the difference between replicates in the phenotypic states of cells, e.g. due to them experiencing different environments, in addition to any potential technical variations in measurements of these states. The method section (page 17, lines 29-34) and the legend of the Dataset EV2 has now been edited to better describe this. To check how this intra-strain biological and technical variance impacts the predictability, we have now compared how well this variance correlates with our capacity to predict individual phenotypes, i.e. with the r^2 scores from phenotypic predictions on the test set and found that the two are weakly anti-correlated Supplementary Figure 7A. This suggests some impact of the intra-strain variance on the predictability of the traits; however, a strong relation between the two is not evident. This is now shown in Supplementary Figure 7A and added in page 8, lines 13-14. We also note that the bulk of our phenotypes capture colony growth on a shared nutrient medium. We have previously in two methods papers explored the nature of between replicate variation for this data type, showing that a) it can be well understood by models of how nutrients are consumed and diffuse across the shared nutrient medium b) and that this explains why it seems dictated by variations in the initial number of cells per colony, and by variations in the distances between colonies. We now refer the readers to these papers in the methods (Zackrisson *et al.*, 2016; Borse *et al.*, 2024).

- Traits may not all show the same degree of variation among strains, which could also contribute to trait predictability. It seems important to show and discuss how trait variation among strains is related to trait predictability.

[R] We note that variance is very similar across traits because the data for most phenotypes was normalized as part of the data pre-processing, meaning that the data values were mathematically adjusted so that phenotypes on different scales should be expressed on a notionally common scale. This dramatically reduces the difference in ranges between phenotypes, and thus also the difference in how much they tend to vary. The predicted r^2 scores vary across phenotypes to a much larger extent than the variance. Thus, we reduced the problem referred to by the reviewers at the data-preprocessing stage. We tried different statistical measures (COV, Relative IQR, and Gini coefficient) to quantify variance among strains, and compared this variance to the predictability of each trait. Regardless of the variance measure, we find very little agreement between trait variance and predictability and note that phenotypes with very similar variance nevertheless vary very substantially in predictions. An example using the COV measurement and predictability is shown in Supplementary Fig. 7B.

However, there is co-variation between trait predictability and trait-type. Notably, sporulation traits and chronological life spans are in general better predicted across conditions (Figure 4D and Supplementary

Fig. 7C). Also, for the growth rate and growth yield measured in the same condition, we observed growth yield to be significantly better predicted compared to growth rate (Supplementary Fig. 7D).

[A] This has now been added in the manuscript at page 8, lines 14-17 along with the addition of Supplementary Fig. 7.

- "The data was scaled using standard scaling" (p21) is too imprecise to understand what was done.

[R&A] We used the `StandardScaler()` function from the `sklearn` library that standardizes the data on a normal scale so that the data has a mean 0 and variance 1. This is done to put all the features on the same scale and avoid having features on widely different scales, which is the requirement for most machine learning methods. This is now explained in page 20, lines 28-30.

- It is unclear what information is used about each predictor for training the model. For example in the case of CNVs, does the model use information about genomic location of CNVs, their length, the number of genes included, the number of copies? Similarly, what information is used for SNPs?

How are LOF variants defined and why do they not overlap with SNPs (as shown in Figure 2a)?

[R&A] The method section has been expanded to further describe the genomic predictors (page 18, lines 12-23).

- Trait-trait correlations are quantified but their statistical significance is not analyzed. The sentence of line 5 of page 4 "we found significant correlations to be both..." is therefore wrong because the authors do not know which correlations are significant. Similarly, no statistical test is provided to support the correlation between azole and CLS traits.

[R] We are sorry that this had not been clearly stated before. However, Figure 1C consisted only of the significant correlations (with $p\text{-value} \leq 0.05$), and the non-significant p -values were represented by blank spaces.

[A] We recalculated all the p -value calculations with correction using False discovery rate (FDR) correction through the Benjamini-Hochberg procedure. Figure 1C is updated with the new plots considering the corrections for multiple testing. The general results remain the same, and we still observe the significant correlations between CLS and azoles and the trade-off between the yield and the growth rate. Also, we clearly state this now in the legend of Figure 1C.

- line 24 of p4 mentions "hierarchical clustering with $k=4$ ": did the authors apply k -means clustering? Or did they cut a hierarchical tree?

[R&A] The number of clusters in the data was calculated using Silhouette's score, which shows how well the cluster has grouped the data. The cluster number with the highest Silhouette scores was 4 clusters. Next, we performed a hierarchical clustering, cutting the tree to obtain 4 clusters. We rephrased the text in the manuscript to make this clearer (page 3, lines 33-34 and page 4, line 1).

- There are several instances where a direct validation of genetic effects is mentioned, without showing the results. AIM29 (line 27, p7): The results are accessible on a GitHub page and mentioned as "manuscript in prep", while simultaneously citing ref 17. What does this mean? Simply show the results in the present manuscript or please don't mention them. SKN7 (p7): where are the results? For SEC11: Supp Data 12 mentions "Unpublished data from <https://www.nature.com/articles/s41559-022-01671-9>". What does this mean? Published or unpublished? Supp Data 14 and 15 contain results (p-values) that cannot be interpreted because what was done from SuppData12 is not written. SPO12 is said to be validated (p14) from ref 17: what validation exactly? Where in ref17? Line 22 p 16 "SNPs... did not show detectable phenotypic effects when reconstructed in different backgrounds". Where is the data from these reconstructions? We simply don't know where to look for the results.

[R&A] We agree that the reporting of the validation experiment was not well organised and opted for reporting all the results here in addition to referring to the original data source (page 5, lines 6-9). The paragraph introducing the AIM29, IME1, and SKN7 results has now been modified and Figure EV1 has been included to show experimental results for these genes. We also added further information in Dataset EV5 to clarify how the validation of SEC11 was performed and that the data is released along with this manuscript.

2. If the authors applied GWAS as they wrote in methods (line 29, p20), then they considered all linkage hits passing a 5% family-wise error rate as being significant and described the properties of these hits as an ensemble of results. This way, multiple testing is corrected for the number of genetic markers, but not for the number of phenotypes, which is large here (n=223). This is a major problem because we do not know what would remain of Fig2 (and related text) if the authors had properly controlled for genome x phenome multiple testing. For example, the Manhattan plot of Fig. 2a shows many dots, even with very weak scores, as if they were all statistically significant (according to the legend). If most hits are false positives, as we suspect, then discussing the absence of patterns (eg. pages 7,8) among them is useless. Also, a large fraction of false positives would be a simple explanation of why "GWAS hits also included equal proportions of synonymous and missense mutations" (page 16).

In addition, the authors seem to be using the term "SNPs" as ML predictors to refer to GWAS hits. If this is the case, then i) many of these hits are of poor quality (non-significant) and therefore discovering that they poorly predict traits is not a finding and ii) these features were selected (by GWAS) on the same dataset as the one used for model training, which is a " post-selection inference" issue (same as mentioned below for LASSO).

[R] We started with the GWAS results of the 99 phenotypes evaluated in De Chiara *et al.*, 2022, and for consistency, we performed the GWAS on the rest of the phenotypes (124) in the same way. Hence, the GWAS was performed independently for each phenotype, with a 5% family-wise error rate being calculated by permuting each phenotype and genetic markers 100 times. This is the gold-standard method for correcting multiple phenotypes comparisons in the GWAS, as also shown in the araGWAS catalog developed using a similar number of genomes (~1k) and phenotypes (167), see Togninalli *et al.*, 2018. We note that the classical statistical tests, such as 'Bonferroni' and 'False Discovery Rate (FDR)', assume independence between different markers (Voicheck and Weigel, 2020; Vorbrugg *et al.*, 2024). This is not true, as many of the genetic markers are usually in linkage disequilibrium. Thus, using e.g. FDR to correct for multiple hypothesis testing, would lead to a very high loss of true positives. We therefore use the

family-wise error rate to adjust for multiple hypothesis testing of many markers, but also now include an FDR to account for the number of phenotypes tested.

[A] To test how the results change after performing corrections for both genomic markers and phenotypes, we combined the p-values of all the 223 independent GWAS analyses and corrected them using FDR. Using a threshold of 0.05, we obtained 887 unique significant GWAS hits (1662 less compared to the previous analysis). Out of the 887 GWAS hits, 709 hits overlap with the GWAS hits from the previous analyses. We further predicted the effect of the variants using a Variant Effect Predictor (VEP) and categorized them according to their consequences, such as synonymous, missense, and loss of function. The general results do not change and there are no significant differences between the p-values distribution among different categories. This is added to the manuscript in the methods section on GWAS (page 19, lines 20-28) along with the Supplementary Fig. 11.

In section 4, using SNPs as predictors refers to the entire set of SNPs (500K after filtering out SNPs present at minor allele frequency of less than 5%). We have now explained the details on this SNP matrix in the methods (page 18, line 19-23). The LASSO feature selection is only done on the training dataset and not on the test set avoiding the " post-selection inference" issue. This is now further clarified in the text (see also reply to point 4).

Using GWAS results to rank the genomic features and further using them for predictions was presented in supplementary note 2. We do agree with the reviewer's comment for this supplementary note and we did acknowledge the overestimation of prediction accuracies due to the same data being used for both GWAS and Machine learning in this case. However, this is a limitation arising due to the population size in our study (~650 euploid/diploid strains used for GWAS) and further reducing the sample size would lead to loss of robustness in our results and hence much bigger datasets are needed to perform this analysis in a non-biased manner. We have removed this note and have expanded on the comparison/similarity between the contributing genomic variants from GWAS and ML in section "Biological interpretation of machine learning predictions" (Page 9).

3. The novel knowledge produced by the study is somewhat difficult to discern. For example, the authors emphasize that machine learning showed that some traits are much better than other features (eg. molecular) to predict phenotypes (page 12). This is largely expected given the strong correlations present in the phenotypic matrix. What exactly was discovered in addition to simple expectation? Page 17 (line 8) writes that the study "suggests that predicting complex traits based on genomic information in natural populations is feasible". This is too vague, readers already know that quantitative genetics is feasible. In addition, other studies used ML to detect rare variants associated with phenotypic variation and should be cited (eg, Petrazzini et al., 2024 <https://doi.org/10.1038/s41588-024-01791-x>).

[R&A] We agree with the reviewer on the need to be more explicit on the novelty of this study. We edited the highlighted concluding sentence (Page 12, lines 1-4) and better positioned our study with respect to the state of the art (also following the suggestions by Reviewer 1, e.g. see page 11, lines 10-17. We clarified that our study provides an in-depth and systematic quantification of ML predictions to quantitative phenotypes. This differentiates our work from studies that applied similar approaches to discrete phenotypes applied to a macro-evolutionary time scale.

We thank the reviewer for pointing out the reference from Petrazzini et al, which we have now added to the discussion. However, Petrazzini et al used ML to refine phenotype classes, which were then used to

run associations. Instead, here we used ML directly to detect rare variants by calculating the feature importance scores. We have run additional analysis (reported in EV5) and further expanded the discussion on this in the relevant text section (Page 12, lines 1-4).

Beyond the novel knowledge highlighted above, we believe that the compendium of phenotypes, their analysis (e.g. correlations) and the derived GWAS catalogue can serve as a useful resource for the yeast genetics community or for researchers interested in exploring similar approaches to the one presented here. Furthermore, the Gen-Phen pipeline provided with the study represents a useful tool for the community that can readily be used to perform similar quantitative predictions without extensive knowledge of the various linear and non-linear ML models. Gen-Phen has many parameters that can be easily defined by users such as the test/train size, cross validation folds, feature selection model, and training model, making it adaptable to different datasets.

4. LASSO-based reduction of features preceding machine learning seems to be applied on the same dataset as downstream model trainings. This is an issue called "post-selection inference" that is known to introduce potential biases in the conclusions. Rather, the authors should at a minima separate the data into training and test sets before LASSO selection. The subject should be mentioned and discussed in the paper. References can be found in Kuchibhotla et al, 2022 (<https://doi.org/10.1146/annurev-statistics-100421-044639>).

[R&A] The dataset is indeed first divided into a training and a testing set in the preprocessing section and the feature selection using LASSO selection is only done on the training dataset, which should avoid the potential bias mentioned by the reviewers. This is depicted in Fig. 3A, in the results (page 6, lines 21-25) and methods section (Page 20, Gen-phen pipeline description).

5. The way prediction accuracy is evaluated is not appropriate. Pearson's correlation coefficients between empirical and predicted values inform on the global ability of a model to predict a set of trait values, but not on how accurate each prediction is. The authors should conduct other performance evaluation and provide other standard metrics such as prediction error.

[R&A] We have now updated all the figures with r^2 scores, which is the more widely used evaluation metric for ML models. However, the two quantities (Pearson's correlation coefficients and r^2 scores) are highly correlated as shown in the plot below, and hence, no significant difference between the interpretations of the results was observed. We also report mean squared error (MSE) in the pipeline, however we use r^2 scores for better interpretability and comparison among the phenotypes.

6. The fact that transcriptomics is a poor predictor would be a very intriguing (and therefore interesting) result if the authors showed that the quality of the transcriptomics dataset is high. Otherwise, poor predictive power may simply result from poor quality.

[R&A] The transcriptomics data have been shown to be high quality and robust as shown in the source publication (Caudal *et al.*, 2024). Each individual transcriptome has at least 1 million mapped reads and independent culture replicates for 29 samples show the data are highly reproducible, with an average correlation of 0.94 between replicates and robust to different sample batches. Therefore, the poor predictions from transcriptomics are genuine and intriguing. Although we lack a clear understanding of why transcriptomics data is a poor predictor, specifically compared to other molecular-level data such as proteomics, studies have shown stronger control of the protein regulation through post-transcriptional buffering compared to mRNA expression which is impacted by transcriptional noise. We added this as discussion in the manuscript (page 11, lines 6-9).

Finally, ML is presented here as if it were a standard, homogeneous approach (e.g. page 3, lines 23-27), but it is an entire scientific field. The authors should specify that they used "some Machine Learning approaches" (not all existing ones) "in a standard procedure". There surely exists other ML methods that will perform better or worse in this context. Also, the "Deep Neural Networks" paragraph on page 23 describes what seems to be a specific type of neural networks, i.e., dense (or fully connected), but a wider range of "deep neural networks" exist and this should be mentioned.

[R&A] We have made the necessary changes to clarify that we used some Machine Learning approaches as a standard benchmarking procedures (page 2, line 28-29). Also, we specified the type of neural networks used in the methods (page 22, line 17-18).

Specific comments:

- abstract: "multiple linear and complex machine learning methods". A linear ML method can be complex and non-linear methods can be simple.

[R&A] We agree with the statement and the sentence is changed accordingly, page 1 line 20.

- abstract: The sentence "We determined gradient boosting machines as best performing in 65% of predictions and pangenome as best predictor" is totally unclear, especially regarding the link between the cited method performance and the biological finding.

[R&A] This sentence is now rephrased and the two findings are now separated (page 1, lines 20-23).

- abstract: "($r = 0.2-0.9$)" r is not defined (and this information is maybe too specific for an abstract).

[R&A] This has now been removed from the abstract and is restricted to the results section (page 6, lines 33-34 and page 7, line 1; of note we now use r^2 scores).

- line12, p4, the letter 'r' is used to mention too different things in the same sentence.

[R&A] This sentence is now rephrased and it is clearly specified what R refers to (page 3, lines 11-13 and 20-23).

- line13, p4: "with that such"?

[R&A] This has been corrected (page 3, lines 20-23).

- line26, p7: "exceptionally high p-values": No, the $-\log_{10}$ of them is high, so the p-values are low.

[R&A] This has been rephrased now, page 5, line 6.

- line 18, p8: PPR is mentioned here, but the methods describe "network expansion" instead. Please be consistent.

[R&A] This has now been changed in the methods, page 20, line 7 and 11 for consistency.

- what does the star on Fig1d represent? How does it "highlight" the connection between phenotypes measured in different studies? (line 17 p 4)?

[R&A] The star in Fig 1D shows the case of growth measured in cycloheximide (CHX) coming from three different studies, being strongly correlated. This is now elaborated in the Fig. 1 legend (page 14, lines 13-14).

- Figure 2b: Numbers above box plots probably correspond to p-values, but this should be made more explicit. How was SNPs weight (y-axis) calculated?

[R&A] This is now specified in the Figure 2B legend (page 14, lines 24-26). The SNP weight or effect size (Beta coefficient) is a statistic obtained from the linear mixed model which quantifies how much a SNP influences the trait.

- line 17, p10: SVR is not defined. Is it the same thing as SVM defined in line 12?

[R&A] SVR is now abbreviated and defined (page 6 line 27) and yes, support vector regressor (SVR) and support vector machine (SVM) were used to refer to the same thing. We now use SVR to be consistent throughout.

- line 28, p10: NN is not defined.

[R&A] This is now abbreviated, page 6 line 27.

- Figure 2c: The plot is difficult to read because phenotypes are only shown on y-axis and not shown on x-axis (Figure 1c is easier to read).

[R&A] An additional colorbar has been added to the x-axis in Figure 2C.

- Figure 3c: Did the authors use data points (black dots) shown on Figure 3c to perform Wilcoxon tests? BayesGBM and BayesSVR do not seem to possibly differ at P-value lower than 0.05.

[R&A] To make the comparisons between the methods clearer, this figure has now been changed with the updated results. Figure 3C depicts the pairwise comparison between few models with Figure EV2 showing the comparisons between all pairwise models and also the comparison between the distributions of predictions using boxplot. The p-values reported are using paired Wilcoxon test which is now specified in the legend.

- Figure 3d: Y-axis label "Fraction of best predicted phenotypes" differs from the legend "Fraction of the best performance...". What exactly is shown? Why is a pattern seen here and not in panel 3c?

[R&A] To rank the methods, we calculated the number of phenotypes that are best predicted by a specific method compared to the rest of the methods divided by the total number of phenotypes. This is defined as the fraction of best-predicted phenotypes. While the initial boxplots in Fig 3C depicted the distributions of the prediction values, which has now been replaced with a pairwise comparison between the methods using scatterplots shown in Fig 3C and Figure EV2A.

- Figure 4a: Axes could be labeled more clearly. Why only 4 comparisons are shown among all possible combinations of 2 predictors?

[R&A] We have now further explained the axis in the Figure 4A legend and added all possible pairwise comparisons as Figure 4B.

- in methods, references should be provided for "PLINK v1.90b6.18", "the online web tool called Variant Effect Predictor (VEP)", "LASSO regression", "scikit-learn", "Hi-LASSO Python library 1.0.6", and all methods: ridge regression, elastic regression, support vector regression.

[R&A] All the references are now added in the manuscript (reference numbers: 69, 70, 72-74, 76-82).

References

Zackrisson, M. *et al.* (2016) 'Scan-o-matic: High-Resolution Microbial Phenomics at a Massive Scale', *G3 Genes/Genomes/Genetics*, 6(9), pp. 3003–3014. Available at: <https://doi.org/10.1534/g3.116.032342>.

Borse, F. *et al.* (2024) 'Quantifying massively parallel microbial growth with spatially mediated interactions', *PLoS computational biology*, 20(7), p. e1011585. Available at: <https://doi.org/10.1371/journal.pcbi.1011585>.

De Chiara, M. *et al.* (2022) 'Domestication reprogrammed the budding yeast life cycle', *Nature Ecology & Evolution*, 6(4), pp. 448–460. Available at: <https://doi.org/10.1038/s41559-022-01671-9>.

Togninalli, M. *et al.* (2018) 'The AraGWAS Catalog: a curated and standardized Arabidopsis thaliana GWAS catalog', *Nucleic Acids Research*, 46(D1), pp. D1150–D1156. Available at: <https://doi.org/10.1093/nar/gkx954>.

Voickek, Y. and Weigel, D. (2020) 'Identifying genetic variants underlying phenotypic variation in plants without complete genomes', *Nature Genetics*, 52(5), pp. 534–540. Available at: <https://doi.org/10.1038/s41588-020-0612-7>.

Vorbrugg, S. *et al.* (2024) 'Gfa2bin enables graph-based GWAS by converting genome graphs to pan-genomic genotypes'. Available at: <https://doi.org/10.1101/2024.12.05.626966>.

Caudal, É. *et al.* (2024) 'Pan-transcriptome reveals a large accessory genome contribution to gene expression variation in yeast', *Nature Genetics*, 56(6), pp. 1278–1287. Available at: <https://doi.org/10.1038/s41588-024-01769-9>.

26th May 2025

Manuscript Number: MSB-2024-12723R

Title: Predicting natural variation in the yeast phenotypic landscape with machine learning

Author: Sakshi Khaiwal

Matteo Chiara

Benjamin Barre

Inigo Barrio-Hernandez

Simon Stenberg

Pedro Beltrao

Jonas Warringer

Gianni Liti

Dear Dr. Liti,

Thank you for submitting the revised version of your manuscript. We have now received feedback from all three reviewers. As you will see from their comments below, Reviewers #1 and #2 are satisfied with the revisions you have made. However, Reviewer #3 has raised substantial concerns and feels that several key issues identified in the previous round remain insufficiently addressed.

Our editorial policy typically allows only one round of major revision. However, we feel that the concerns raised by Reviewer #3 are potentially addressable, and given the balance of these evaluations from all three reviewers, we are willing to consider another revision of your manuscript, provided you can thoroughly respond to the points raised by Reviewer #3.

Please note that the revised submission will be subject to further review, and we cannot guarantee acceptance at this stage. If you choose to proceed, we ask that you include a detailed, point-by-point response to Reviewer #3's comments with your revised manuscript.

On a more editorial level:

1. Please provide up to five keywords.
2. Remove "Authors' contribution" section from the manuscript.
3. For the next submission, please include the institutional email address for the co-corresponding author, Sakshi Khaiwal.
4. Ensure that the funding information in the submission system matches the information provided in the manuscript file. The following funding information is missing in the submission system - Agence Nationale de la Recherche (ANR-11-LABX-0028-01, ANR-18-CE12-0004, ANR-20-CE12-0020, ANR-22-CE12-0015, ANR-24-CE12-7740), Fondation pour la Recherche Médicale (EQU202003010413), Impulscience - Fondation Bettencourt Schueller, UCA AAP Start-up Deep tech, CEFIPRA, SATT Sud-Est; European Union's Horizon 2020 research and innovation programme under grant agreement No 847581, the Region Sud, and the UCA J.E.D.I
5. The references need to be formatted according to the Molecular Systems Biology reference style. Please list up to 10 co-authors of a paper before adding et al. in the reference list. Citations should be listed in alphabetical order.
6. "Code Availability" should be combined with "Data Availability" and the heading should be removed.
7. "Competing interest" should be renamed to "DISCLOSURE AND COMPETING INTERESTS STATEMENT".
8. Appendix File
 - The appendix must be in PDF format.
 - The title page should include the phrase: "Appendix for [Manuscript Title]".
 - References within the appendix should be listed in alphabetical order, using the format of 10 authors followed by "et al."
 - Figures and tables should be labeled and cited as "Appendix Figure Sx" and "Appendix Table Sx" respectively. Do not refer to them as "Supplementary Figures/Tables".
9. Callouts
 - Callouts for Appendix Figures S12-S15 are missing and should be included in the manuscript.
 - There are callouts for 9 datasets, but none have been uploaded. This discrepancy must be addressed.

10. Please download and fill our Reagents and Tools Table template (.docx), which you can find in our author guidelines: <https://www.embopress.org/page/journal/17444292/authorguide#structuredmethods>.

11. Please address the following issues in figure legends:

- Please indicate the statistical test used for data analysis in the legends of figures 1C, 2A, 3B,C; 4A, B, E; EV2 A.
- Please note that the box plots need to be defined in terms of minima, maxima, centre, bounds of box and whiskers, and percentile in the legends of figures 2B, 4A, E; EV1 A, B; EV2 B
- Please note that information related to n is missing in the legends of figures 2B, 4A, E; EV1 A, B; EV2 B
- Please note that the error bars are not defined in the legend of figure EV1 A

12. "Main" should be renamed to "Introduction" and "Material and Methods" to "Methods"

13. Section order should be corrected to : Title page - Abstract & Keywords - Introduction - Results - Discussion - Methods - Data Availability - Acknowledgements - Disclosure and Competing Interests Statement - References - Figure Legends - Table(s) - Expanded View Figure Legends.

When you resubmit your manuscript, please download our CHECKLIST (<https://bit.ly/EMBOPressAuthorChecklist>) and include the completed form in your submission.

Please note that the Author Checklist will be published alongside the paper as part of the transparent process (<https://www.embopress.org/page/journal/17444292/authorguide#transparentprocess>).

If you feel you can satisfactorily deal with these points and those listed by the referees, you may wish to submit a revised version of your manuscript. Please attach a covering letter giving details of the way in which you have handled each of the points raised by the referees. A revised manuscript will be once again subject to review and you probably understand that we can give you no guarantee at this stage that the eventual outcome will be favorable.

We look forward to receiving your revised manuscript soon.

Kind regards,
Jingyi

Jingyi Hou, PhD
Senior Editor
Molecular Systems Biology

We realize that it is difficult to revise to a specific deadline. In the interest of protecting the conceptual advance provided by the work, we recommend a revision within 3 months (24th Aug 2025). Please discuss the revision progress ahead of this time with the editor if you require more time to complete the revisions.

*** PLEASE NOTE *** As part of the EMBO Press transparent editorial process initiative (see our Editorial at <https://dx.doi.org/10.1038/msb.2010.72>), Molecular Systems Biology publishes online a Review Process File with each accepted manuscripts. This file will be published in conjunction with your paper and will include the anonymous referee reports, your point-by-point response and all pertinent correspondence relating to the manuscript. If you do NOT want this File to be published, please inform the editorial office at contact@molsystbiol.org within 14 days upon receipt of the present letter.

Reviewer #1:

The authors have done a great job addressing my comments and suggestions. I have no further comments.

Reviewer #2:

We thank the authors for answering all our queries and implementing all suggestions. We believe that this revised version further improves what essentially already was an impressive study that will be of interest to the broad community of colleagues investigating the use of machine learning to analyze the genotype-to-phenotype connection in different organisms. We do not have any further remarks.

Dimitrios Konstantinidis, Michiel Schreurs & Kevin J. Verstrepen

Reviewer #3:

The manuscript quality was improved by the revisions. However, the issues noted previously have been only partially addressed.

Regarding major point 1, the revised methods and text now provide better precision and details on what was done, but problems persist:

- We criticized the absence of results describing the validation experiments on causal effects of genes AIM29, SEC11 and SPO12. The revised manuscript now clearly describes the validation experiment for another gene, IME1: Fig. EV1A is a reproduction of Extended Data Fig. 3 from a previous paper of this laboratory (ref 20). The novel results on AIM29 are now shown on Figure EV1B, from data made available via github. However, no annotation is provided on this git repository (strains are "BY" and "N13" and do not match the figure legend) and how the strains were generated and validated is not mentioned. This does not align with standard practices in publishing yeast genetics results. Regarding SEC11, an allele-swapping experiment is mentioned but still, no figure is presented. The text refers to "Dataset EV5" which was not made available for review so it is not possible to evaluate this revision. Regarding SPO12, it is now clearer how the authors claim validation: they write on page 9 "SPO12 was experimentally validated to determine dyads formation (ref20)" and repeat this assertion on page 12 line 2. This is problematic: in Ref 20, which largely uses the same data as here, this lab applied a candidate-gene approach on SPO genes that escaped GWAS-detection. They "found these to be disproportionately likely to carry variants predicted to result in loss-of-function in domesticated lineages but more rarely contained such variants in wild strains....". Extended Data Fig.4 of ref 20 shows a zoom on the genotypes and phenotypes of natural strains, but does not show any validation experiment. I could not find a validation experiment elsewhere in this publication. So, either I missed it in ref20, or- if such validation is indeed absent in ref20, then referencing it as evidence is misleading. The authors should clarify this.
- The added analysis of intra-strain and inter-strain trait variation is useful. However, axis-labeling on Supplementary Fig. 7 is confusing: The y-axis label on S7A ("CV score") is not mathematically defined, and differs from "r²" indicated in the legend, or is it "P(AF)" mentioned at the top? X-axis "mean replicates variance" is not explicit and should be changed for "Mean intra-strain variance" or similar. Same for S7B: "COV" is on the figure and "trait variance at the population level" is in the legend, these are different metrics.
- Text on clustering method (p3 lines 33-34) improved but remains imprecise. Silhouette scores are computed after clustering. Text or methods should explicitly mention that hierarchical clustering was performed first and was followed by silhouette analysis.

Regarding major point 2, the authors did not address correctly the issue on GWAS significance in the context of numerous traits. Figure 2 and related text therefore remain unreliable.

- The authors wrongly write that performing GWAS independently on each phenotype is a "gold-standard" in the case of multiple (numerous) traits. Gold-standards have been thoroughly established in the case of expression traits (eQTL) which are a typical case of multiplicity. For example, in ref Caudal 2024 (cited here as the source of expression traits), as in many eQTL studies, authors applied GWAS independently on each trait and then appropriately controlled for the FDR (Benjamini-Hochberg); this is not done here.
- The authors write that "the classical tests, such as Bonferroni and False Discovery Rate (FDR), assume independence between different markers". First, these methods are not tests but analyses of the outcome (p-values) of tests. More importantly, it is incorrect to say that FDR assumes independence between the tests. FDR can be estimated (and therefore controlled) by assuming a uniform distribution of the p-values under consideration; without assuming independence between the tests. It is nonetheless true that cases of dependencies exist where the distribution is non-uniform. In such situations, the FDR must be controlled differently, for example via empirical permutations.
- As a support to their approach, the authors cite 3 references in their response and revised text. Togninalli 2018, who describe araGWAS: a database and web interface allowing users to explore genetic linkages of their favorite plant trait(s). It is true that Togninalli et al. applied GWAS on a large number of traits. They performed permutations to determine, for every trait, a 5% family-wise error rate and retain corresponding linkage hits. The corresponding p-values are made available per-trait, with absolutely no consideration on the number of traits studied. This is perfectly acceptable when producing a database to the community, as in Togninalli 2018, because it is then the responsibility of each user to control for the number of traits he/she studies. But focusing on single-traits p-values is not acceptable when interpreting global lists of hits, as done here. Citing the

araGWAS reference is therefore inappropriate regarding the issue discussed here. The second reference cited is Voichek & Weigel 2020, who developed a k-mer based method to perform linkage using incomplete genomes. I could not find any mention of the FDR in this paper. The third citation is a pre-print from Vorbrugg et al. 2024 describing a method to perform GWAS using variation graphs. Again, there is no mention of the FDR in this manuscript.

- The authors nonetheless started to address the issue. They describe (in methods) an analysis where they "combined the p-values of all the 223 independent GWAS analyses and corrected them using FDR". Results are shown in Supp Fig S11. How p-values were "combined" is not described. One can assume they used the method of Storey & Tibshirani PNAS 2003. In this case, was the distribution of p-values uniform? FDR control consists in selecting a set of preexisting p-values using a whole-analysis cutoff. In this case, the reported p-values in FigS11 differ from the original ones of Fig2. What are these "novel" p-values? They might be q-values as in Storey 2003. but there is no way to be sure. It appears two of the highest hits of Fig2 (SKN7 and AIM29) are lost in Fig S11; and the highest hit on FigS11 (chr13) is not noticeable on Fig2. Can the authors provide explanations to this?

- Could the authors please explain why Fig2A was revised by simply changing the bottom labeling of the y-axis? The dots seem to be the same. So, the bottom dots that were initially extremely close to zero are now at ~3.9. The authors do not explain this change in their response.

Other major points were satisfyingly addressed.

Specific comments were satisfyingly addressed except the following ones:

- Initial Fig. 3c is now Fig. EV2C and reports a Wilcoxon-test p-value of $3.2e-5$ between BayesGBM and BayesSVR data points. Given the distributions of these data points, I believe there is an error with the pairwise tests applied on this figure. The editor could request the source data of this figure for review.

- Figure 3D and EV2A: "Fraction of predicted phenotypes" is defined in the response to reviewers but still not in legend nor in methods.

Point-by-point response to Reviewer

Please find below are our replies in blue to the remaining points [R: reply, A: action]

Reviewer #1:

The authors have done a great job addressing my comments and suggestions. I have no further comments.

Reviewer #2:

We thank the authors for answering all our queries and implementing all suggestions. We believe that this revised version further improves what essentially already was an impressive study that will be of interest to the broad community of colleagues investigating the use of machine learning to analyze the genotype-to-phenotype connection in different organisms. We do not have any further remarks.
Dimitrios Konstantinidis, Michiel Schreurs & Kevin J. Verstrepen

Reviewer #3:

The manuscript quality was improved by the revisions. However, the issues noted previously have been only partially addressed.

Regarding major point 1, the revised methods and text now provide better precision and details on what was done, but problems persist:

- We criticized the absence of results describing the validation experiments on causal effects of genes AIM29, SEC11 and SPO12. The revised manuscript now clearly describes the validation experiment for another gene, IME1: Fig. EV1A is a reproduction of Extended Data Fig. 3 from a previous paper of this laboratory (ref 20). The novel results on AIM29 are now shown on Figure EV1B, from data made available via github. However, no annotation is provided on this git repository (strains are "BY" and "N13" and do not match the figure legend) and how the strains were generated and validated is not mentioned. This does not align with standard practices in publishing yeast genetics results. Regarding SEC11, an allele-swapping experiment is mentioned but still, no figure is presented. The text refers to "Dataset EV5" which was not made available for review so it is not possible to evaluate this revision. Regarding SPO12, it is now clearer how the authors claim validation: they write on page 9 "SPO12 was experimentally validated to determine dyads formation (ref20)" and repeat this assertion on page 12 line 2. This is problematic: in Ref 20, which largely uses the same data as here, this lab

applied a candidate-gene approach on SPO genes that escaped GWAS-detection. They "found these to be disproportionately likely to carry variants predicted to result in loss-of-function in domesticated lineages but more rarely contained such variants in wild strains....". Extended Data Fig.4 of ref 20 shows a zoom on the genotypes and phenotypes of natural strains, but does not show any validation experiment. I could not find a validation experiment elsewhere in this publication. So, either I missed it in ref20, or- if such validation is indeed absent in ref20, then referencing it as evidence is misleading. The authors should clarify this.

[R & A] We now further clarify and add additional details as suggested by the reviewer. Specifically:

AIM29 - The polymorphism was engineered in the BY4743 (lab strain) genetic background. The engineered strain is labelled as 'N13', and multiple replicates of both the wild-type strain and the engineered strain were phenotyped for growth in ketoconazole. We added additional information on the experimental validation, strain construction, and nomenclature in the data files under the same GitHub link (https://github.com/SakshiKhaiwal/CEFIPRA/blob/main/Data/AIM29_validation_GT.xlsx https://github.com/SakshiKhaiwal/CEFIPRA/blob/main/Data/Aim29_validation_Yield.xlsx).

SEC11- We apologise for the problem of the missing expanded datasets, which have now been uploaded. Dataset EV5 contains all the information regarding the SEC11 phenotype experiment.

SPO12- We apologise for the use of 'experimentally validated", which is not the correct wording. Indeed, the SPO12 data were obtained from ref. 20. Conclusions were based on:

- a) the strong loss-of-function mutations (a frameshift and a premature stop codon in the central part of the gene) observed
- b) the severe meiosis phenotypes of gene deletion strains lacking a functional SPO12 (such mutants only complete a single meiotic division)
- c), the fact that the mutations observed produce 100% of dyads, i.e., only two out of four gametes are formed.

The text has now been modified (page 9, lines 24-28 and page 12, lines 12-15) to clarify this aspect.

- The added analysis of intra-strain and inter-strain trait variation is useful. However, axis-labeling on Supplementary Fig. 7 is confusing: The y-axis label on S7A ("CV score") is not mathematically defined, and differs from "r2" indicated in the legend, or is it "P(AF)" mentioned at the top? X-axis "mean replicates variance" is not explicit and should be changed for "Mean intra-strain variance" or similar. Same for

S7B: "COV" is on the figure, and "trait variance at the population level" is in the legend, these are different metrics.

[R & A] The definition of "CV scores" has now been added to the legend of Appendix Figure S7 to clarify that it is a mean cross-validated R² score from prediction. The x-axis of Figure S7A has now been changed to "Mean intra-strain variance". Also, the legend of Figure S7B now explicitly describes the COV.

- Text on clustering method (p3 lines 33-34) improved but remains imprecise. Silhouette scores are computed after clustering. Text or methods should explicitly mention that hierarchical clustering was performed first and was followed by silhouette analysis.

[R & A] The two calculations, hierarchical clustering and Silhouette scores, are independent. We now specify that the optimal number of clusters estimated from Silhouettes' scores using k-means clustering with k ranging from 1-20 was used to cut the hierarchical tree (Page 4, lines 4-7).

Regarding major point 2, the authors did not address correctly the issue on GWAS significance in the context of numerous traits. Figure 2 and related text therefore remain unreliable.

- The authors wrongly write that performing GWAS independently on each phenotype is a "gold-standard" in the case of multiple (numerous) traits. Gold-standards have been thoroughly established in the case of expression traits (eQTL) which are a typical case of multiplicity. For example, in ref Caudal 2024 (cited here as the source of expression traits), as in many eQTL studies, authors applied GWAS independently on each trait and then appropriately controlled for the FDR (Benjamini-Hochberg); this is not done here.

[R] We respectfully believe there may have been a misinterpretation of our previous response. We would like to clarify that the use of empirical simulations to control the family-wise error rate (FWER) for every analyzed trait individually is a widely accepted and standard practice in the GWAS field (Gao *et al*, 2010; John *et al*, 2024).

It is important to distinguish between controlling the FWER and the false discovery rate (FDR), as these represent two distinct approaches to managing multiple testing errors. While classical procedures such as the Bonferroni correction can be used to control FWER, it is well-documented that these methods tend to be overly conservative in the context of genome-wide association studies (GWAS), where millions of tests are performed (Asif *et al*, 2021; Johnson *et al*, 2010). To address this, a more powerful and commonly adopted strategy involves the use of empirical permutations to estimate significance thresholds, thereby controlling the FWER at 5%. In this framework, the probability of observing one or more false positives per

trait is maintained at, or below 5%, which gives a reasonable trade-off between type I and type II errors. This empirical approach is widely regarded as the standard for controlling for false positives in GWAS analyses, without failing to call an excessive number of true positives. For further context, we kindly refer the reviewer to the following references, where similar approaches have been applied to GWAS across multiple phenotypes: (Peter *et al*, 2018; Togninalli *et al*, 2018; De Chiara *et al*, 2022; Voichek & Weigel, 2020; Vorbrugg *et al*, 2024; Caudal *et al*, 2024).

The reviewer specifically referenced Caudal *et al.* (2024) on how to appropriately solve this problem by applying GWAS independently on each trait and then controlling for the FDR (Benjamini-Hochberg). However, the reviewer's interpretation of the method used in this paper is incorrect. Caudal *et al.* conducted GWAS on 6,119 gene expression traits and used a 5% family-wise error rate (FWER) to determine trait-specific significance thresholds, which is also the approach we adopted in our work. Here is the exact text written by Caudal *et al*:

“A trait-specific P threshold was established for each gene by permuting phenotypic values between individuals 100 times. The significance threshold was the 5% quantile (the fifth lowest P value from the permutations) in each set, which was then Benjamini–Hochberg-adjusted to account for multiple test bias.”

[Caudal *et al.* Page 13, Methods section GWAS].

Since the GWAS on CNV and SNPs were run separately, the BH method was simply applied to adjust the significance thresholds to correct for the difference in number between CNV and SNPs (leading to higher p-value thresholds for CNV and lower for SNPs). To clarify this point, we contacted the corresponding authors of the Caudal *et al.* study (full exchange of correspondence is now provided to the editor). The authors stated:

“adjustments were made phenotype by phenotype, but none have been made to account for the total number of phenotypes.”

We emphasize that, as is standard practice for multiple-trait GWAS, we have controlled for the FWER per trait. To confirm that this leads to an overall low rate of false discoveries, we provide an estimation of the FDR for 223 phenotypes.

[A] We have now estimated the FDR in our global GWAS analysis. For GWAS at the single-trait level, we used the classical permutation-based approach to account for complex confounding factors, notably many variants being in linkage disequilibrium (LD) and thus not being independently tested for in other tests. For each trait, we ran 100 permutations, i.e., the phenotypes were reshuffled 100 times across variants, and the GWAS was run for each case. The 5th smallest p-value out of these 100 analyses was used as the significance threshold. Associations

below this threshold were considered significant associations and termed 'GWAS hits'. This method controls the FWER at 5%, i.e., the chance of having at least one false positive under the null hypothesis is less than or equal to 5%. Since we ran 223 GWAS analyses, each controlled at FWER at 5%, we can approximate with the number of false positives as at least $= 223 * 0.05 = \sim 11$. As there is no straightforward way to quantitatively specify what "at least" translates into, this ~ 11 estimation should be seen as an approximation. Given that the estimated FDR = expected false positives/rejected hypotheses (GWAS hits) and we have 4846 rejected hypotheses, we estimate an $FDR = 11/4846 = 0.002476269$

The FDR is at $\sim 0.25\%$, which is quite low as compared to the commonly accepted threshold of 5%. Hence, we do not expect our overall conclusions from GWAS hits to be meaningfully biased by false positives, although some individual hits, of course, are expected to be false calls. We hope this clarification is helpful and confirms that our approach is consistent with standard practices in the field.

- The authors write that "the classical tests, such as Bonferroni and False Discovery Rate (FDR), assume independence between different markers". First, these methods are not tests but analyses of the outcome (p-values) of tests. More importantly, it is incorrect to say that FDR assumes independence between the tests. FDR can be estimated (and therefore controlled) by assuming a uniform distribution of the p-values under consideration, without assuming independence between the tests. It is nonetheless true that cases of dependencies exist where the distribution is non-uniform. In such situations, the FDR must be controlled differently, for example, via empirical permutations.

[R] We would like to differentiate between the two mentioned methods. Bonferroni is used to control the FWER; however, using it for a large set of tests leads to overly conservative estimates, and most of the true significant signals are lost (Johnson *et al*, 2010; Kaler & Purcell, 2019; Gao *et al*, 2010). Hence, using empirical permutations to control the FWER is a more appropriate method.

Benjamini–Hochberg (BH) is an FDR-based correction which indeed does assume independence or positive dependence among the test statistics (Benjamini & Hochberg, 1995; Kaler & Purcell, 2019; John *et al*, 2024), and also has a high level of false negatives. BH is therefore not suitable in GWAS analyses where the purpose is to catalogue candidate GWAS hits, and where it is important not to overlook the majority of true associations, and where false positives can be filtered out in a more downstream analysis (Schwartzman & Lin, 2011; Kaler & Purcell, 2019; John *et al*, 2024). Moreover, different traits have different genetic structure and heritability; hence, the significance threshold should be estimated at the trait level (Kaler & Purcell, 2019).

Moreover, we see very strong correlations among the phenotypes. Indeed, from our PCA, we found that the first two PCs explain 70% of the variance, while more than

95% of the variance is explained by only 15 PCs. This suggests that a relatively small number of highly pleiotropic variants explain much of the total phenotypic variation. This dependence across phenotypes is very hard to account for statistically, and our test, which assumes independent variant effects across phenotypes, is conservative - i.e., the true number of false positives is likely to be lower than we estimate above.

We have thoroughly searched the literature for the cases where the two types of error control (FWER and FDR) are used together; most studies either use a constant threshold (Canela-Xandri *et al*, 2018; Karczewski *et al*, 2022; Galardini *et al*, 2019) or perform empirical simulations to estimate significant thresholds per trait (Togninalli *et al*, 2018; Peter *et al*, 2018; De Chiara *et al*, 2022; Voichek & Weigel, 2020; Vorbrugg *et al*, 2024; Caudal *et al*, 2024).

- As a support to their approach, the authors cite 3 references in their response and revised text. Togninalli 2018, who describe araGWAS: a database and web interface allowing users to explore genetic linkages of their favorite plant trait(s). It is true that Togninalli *et al*. applied GWAS on a large number of traits. They performed permutations to determine, for every trait, a 5% family-wise error rate and retain corresponding linkage hits. The corresponding p-values are made available per-trait, with absolutely no consideration on the number of traits studied. This is perfectly acceptable when producing a database to the community, as in Togninalli 2018, because it is then the responsibility of each user to control for the number of traits he/she studies. But focusing on single-traits p-values is not acceptable when interpreting global lists of hits, as done here. Citing the araGWAS reference is therefore inappropriate regarding the issue discussed here. The second reference cited is Voichek & Weigel 2020, who developed a k-mer based method to perform linkage using incomplete genomes. I could not find any mention of the FDR in this paper. The third citation is a pre-print from Vorbrugg *et al*. 2024 describing a method to perform GWAS using variation graphs. Again, there is no mention of the FDR in this manuscript.

[R] The main motivation for performing GWAS in our case was indeed to provide a yeast GWAS catalogue for the community (please see Dataset EV4), as in the case of Togninalli *et al*. (2018). Voichek & Weigel 2020 was presented as a reference, as the authors have used a similar GWAS approach. They used k-mers and SNPs as the genotype data and analysed 1,582 *A. thaliana* phenotypes, 252 maize traits, and 96 tomato traits. The permutation-based approach was used to set the thresholds to identify significantly associated k-mers and SNPs; however, no further FDR corrections were performed on the phenotypes. Also, more downstream analysis with the GWAS hits from all the traits combined has been shown for all three cases.

Similarly, in the preprint Vorbrugg *et al*. 2024, where variation graphs are integrated with GWAS on 1695 *A. thaliana* traits, the intention was to highlight that multiple testing is performed on the genotype (graph-nodes in this case) using the

permutation-based approach, and no further FDR correction is performed on the thresholds. However, the overlapping significant hits from the graph-nodes, SNPs, and k-mers are compared for the 1695 *A. thaliana* phenotypes.

To our knowledge, no existing framework jointly controls both the family-wise error rate (FWER) and the false discovery rate (FDR) across genotypes and phenotypes in GWAS. After a thorough review of the GWAS literature, we conclude that the most effective strategy currently available for controlling false positives is the permutation-based FWER approach.

- The authors nonetheless started to address the issue. They describe (in methods) an analysis where they "combined the p-values of all the 223 independent GWAS analyses and corrected them using FDR". Results are shown in Supp Fig S11. How p-values were "combined" is not described. One can assume they used the method of Storey & Tibshirani PNAS 2003. In this case, was the distribution of p-values uniform? FDR control consists in selecting a set of preexisting p-values using a whole-analysis cutoff. In this case, the reported p-values in FigS11 differ from the original ones of Fig2. What are these "novel" p-values? They might be q-values as in Storey 2003. but there is no way to be sure. It appears two of the highest hits of Fig2 (SKN7 and AIM29) are lost in Fig S11; and the highest hit on FigS11 (chr13) is not noticeable on Fig2. Can the authors provide explanations to this?

[R] In response to the reviewers' comment, we explored an alternative scenario in which false discovery rate (FDR) correction was performed jointly across all genotypes and phenotypes. Specifically, we applied the Benjamini-Hochberg (BH) procedure to approximately 20 million p-values (223 phenotypes \times 100,000 genotypes) and obtained GWAS hits using 0.05 as the significance threshold. As the reviewer noted, the resulting adjusted p-values are indeed q-values; we have updated the terminology accordingly in Figure S11 for clarity.

We also evaluated Storey's method, which attempts to adaptively estimate the proportion of true null hypotheses to better account for the effective number of independent tests. However, in our dataset, Storey's method estimated the proportion of independent tests to be 1, effectively treating all tests as independent. This result is inconsistent with known patterns of linkage disequilibrium and phenotype correlation, and thus appears to reflect a limitation of the method in this context. Consequently, the q-values obtained using Storey's approach were effectively identical to those from the BH correction.

The aim of our FDR analysis was to demonstrate that performing FDR correction across all genotypes and phenotypes can substantially increase type II error rates (false negatives). This leads to the loss of top-ranked significant associations and a general reduction in GWAS power. Notably, this approach failed to detect the experimentally validated association in *AIM29*. These results highlight the importance of accounting for the unique genetic structure for each phenotype and underscore the necessity of estimating the significant threshold on a per-phenotype basis to ensure robust detection of true genetic associations.

[A] We have revised the methods section, specifically under “GWAS Error Rates and Limitations,” to reflect the reviewer’s feedback, page 15, lines 5-34 and page 16, lines 1-4. The revised text clarifies the limitations of GWAS in drawing conclusions from individual candidate associations and emphasizes that our GWAS catalog should be viewed as a valuable resource of potential candidate loci for the yeast research community. Furthermore, we highlight that in the context of constructing a comprehensive GWAS catalog and facilitating comparisons with machine learning approaches, minimizing false negatives is a greater concern than minimizing false positives.

- Could the authors please explain why Fig2A was revised by simply changing the bottom labeling of the y-axis? The dots seem to be the same. So, the bottom dots that were initially extremely close to zero are now at ~3.9. The authors do not explain this change in their response.

[R & A] In the previous Fig 2A, the manhattan plot generated using qqman library, we had used the scale break (‘\’) to compress the interval between 0-5 for better visualization to avoid a big gap between 0- ~4 as most bottom dots are above ~4. To generate the updated plot, we used the ‘Topr’ library to make some visual changes suggested by reviewer 2, the manhattan function from this library automatically compress the scale between 0-5 for better visualization and correctly represents the bottom dots at ~3.9 as interpreted by the reviewer and we no longer needed the scale break.

Other major points were satisfyingly addressed.

Specific comments were satisfyingly addressed, except the following ones:

- Initial Fig. 3c is now Fig. EV2C and reports a Wilcoxon-test p-value of $3.2e-5$ between BayesGBM and BayesSVR data points. Given the distributions of these data points, I believe there is an error with the pairwise tests applied on this figure. The editor could request the source data of this figure for review.

[R & A] The plot has been generated using the ggplot2 function with the p-values reported using the ‘stat_compare_means()’ function with method = “wilcox.test”, paired=T and p.adjust.method = “BH”. We repeated our analysis and confirmed that there is no error in the p-values reported. The requested source data used to generate it are provided (uploaded in the “Figure Source Data” category). An all vs all comparison plot from this is shown below.

- Figure 3D and EV2A: "Fraction of predicted phenotypes" is defined in the response to reviewers but still not in legend nor in methods.

[R & A] The explanation of the "Fraction of predicted phenotypes" has been added to the figure legends 3D and EV2A.

References

Asif H, Alliey-Rodriguez N, Keedy S, Tamminga CA, Sweeney JA, Pearlson G, Clementz BA, Keshavan MS, Buckley P, Liu C, *et al* (2021) GWAS significance thresholds for deep phenotyping studies can depend upon minor allele frequencies and sample size. *Mol Psychiatry* 26: 2048–2055

Benjamini Y & Hochberg Y (1995) Controlling the False Discovery Rate: A Practical and

- Powerful Approach to Multiple Testing. *J R Stat Soc Ser B Stat Methodol* 57: 289–300
- Canela-Xandri O, Rawlik K & Tenesa A (2018) An atlas of genetic associations in UK Biobank. *Nat Genet* 50: 1593–1599
- Galardini M, Busby BP, Vieitez C, Dunham AS, Typas A & Beltrao P (2019) The impact of the genetic background on gene deletion phenotypes in *Saccharomyces cerevisiae*. *Mol Syst Biol* 15: e8831
- Gao X, Becker LC, Becker DM, Starmer JD & Province MA (2010) Avoiding the high Bonferroni penalty in genome-wide association studies. *Genet Epidemiol* 34: 100–105
- John M, Korte A & Grimm DG (2024) The benefits of permutation-based genome-wide association studies. *J Exp Bot* 75: 5377–5389
- Johnson RC, Nelson GW, Troyer JL, Lautenberger JA, Kessing BD, Winkler CA & O'Brien SJ (2010) Accounting for multiple comparisons in a genome-wide association study (GWAS). *BMC Genomics* 11: 724
- Kaler AS & Purcell LC (2019) Estimation of a significance threshold for genome-wide association studies. *BMC Genomics* 20: 618
- Karczewski KJ, Solomonson M, Chao KR, Goodrich JK, Tiao G, Lu W, Riley-Gillis BM, Tsai EA, Kim HI, Zheng X, *et al* (2022) Systematic single-variant and gene-based association testing of thousands of phenotypes in 394,841 UK Biobank exomes. *Cell Genomics* 2: 100168
- Peter J, De Chiara M, Friedrich A, Yue J-X, Pflieger D, Bergström A, Sigwalt A, Barre B, Freel K, Llored A, *et al* (2018) Genome evolution across 1,011 *Saccharomyces cerevisiae* isolates. *Nature* 556: 339–344
- Schwartzman A & Lin X (2011) The effect of correlation in false discovery rate estimation. *Biometrika* 98: 199–214
- Voichek Y & Weigel D (2020) Identifying genetic variants underlying phenotypic variation in plants without complete genomes. *Nat Genet* 52: 534–540
- Vorbrugg S, Bezrukov I, Bao Z, Xian W & Weigel D (2024) Gfa2bin enables graph-based GWAS by converting genome graphs to pan-genomic genotypes. doi:10.1101/2024.12.05.626966 [PREPRINT]

16th Jul 2025

Manuscript number: MSB-2024-12723RR

Title: Predicting natural variation in the yeast phenotypic landscape with machine learning

Dear Dr. Liti,

Thank you once again for submitting the revised version of your manuscript. We have now received the reviewer's feedback, and as you can see below, they are satisfied with the revisions. I am pleased to inform you that your paper has been accepted for publication.

We also appreciate the thorough and thoughtful responses you provided to the reviewers' comments. It has been a pleasure working with you throughout this process to bring your paper to publication.

Please note that the reviewer made a minor additional comment regarding the improvement of a figure. While this does not affect the acceptance decision, we encourage you to consider the suggestion. If you choose to revise the figure, please feel free to send it by replying to this email, and we will upload the updated version on your behalf.

Kind regards,
Jingyi

Jingyi Hou, PhD
Senior Editor
Molecular Systems Biology

Reviewer #3:

With this last round of revisions, the authors have now addressed the points raised in my previous comments.

In particular:

- I thank the authors for their detailed response to the FWER/FDR issue. Their revisions of the manuscript's methods section now appropriately explain what was done, and readers will have a clearer understanding of the meaning of the significance scores based on per-trait tests. It is helpful that the authors discussed directly with the authors of the Caudal et al. eQTL study. Indeed, their writing of "The significance threshold was the 5% quantile ... in each set, which WAS THEN Benjamini-Hochberg-adjusted ..." suggested that they applied a multiple-traits correction after trait-by-trait GWAS scans. I was therefore surprised that they did not.

For information regarding standards for multiple traits corrections in eQTL studies, the authors can read Huang et al. 2018 (doi.org/10.1093/nar/gky780) who specifically evaluated methods and their power in the case of cis-eQTL mapping (although not GWAS, in species with large genomes such as humans, many SNPs are scanned for each expression trait).

- I also thank the authors for providing the source data of Figure EV2C and explaining their method. Applying Wilcoxon paired test on this data confirms the authors' results, for example:

```
> x <- read.table("MSB-2024-12723RR-Source_data_Fig_EV2C-sd.csv", sep = ",", header = TRUE)
```

```
> wilcox.test(x$BayesGBM, x$BayesSVR, paired = TRUE)
```

Wilcoxon signed rank exact test

data: x\$BayesGBM and x\$BayesSVR

V = 325, p-value = 3.186e-05

alternative hypothesis: true location shift is not equal to 0

My doubts came from the fact that neither the figure nor the legend indicated the paired nature of the data. An unpaired test (which, I agree, would not be appropriate here) would obviously not be as significant ($p = 0.037$ in this case). The revised legend is now correct as it explicitly mentions pairing. The figure might be further improved by drawing lines between dots for some of the pairs, or by using colors, so that pairing is not only mentioned but also visualized.

- all other points were satisfyingly addressed.
